# Don't Pour Cereal into Coffee: Differentiable Temporal Logic for Temporal Action Segmentation

**Ziwei Xu**[†*]  **Yogesh S Rawat**[§]  **Yongkang Wong**[†]  **Mohan S Kankanhalli**[†]  **Mubarak Shah**[§]

† School of Computing, National University of Singapore

§ Center for Research in Computer Vision, University of Central Florida

{ziwei-xu,mohan}@comp.nus.edu.sg    yongkang.wong@nus.edu.sg

{yogesh,shah}@crcv.ucf.edu

## Abstract

We propose Differentiable Temporal Logic (DTL), a model-agnostic framework that introduces temporal constraints to deep networks. DTL treats the outputs of a network as a truth assignment of a temporal logic formula, and computes a temporal logic loss reflecting the consistency between the output and the constraints. We propose a comprehensive set of constraints, which are implicit in data annotations, and incorporate them with deep networks via DTL. We evaluate the effectiveness of DTL on the temporal action segmentation task and observe improved performance and reduced logical errors in the output of different task models. Furthermore, we provide an extensive analysis to visualize the desirable effects of DTL.

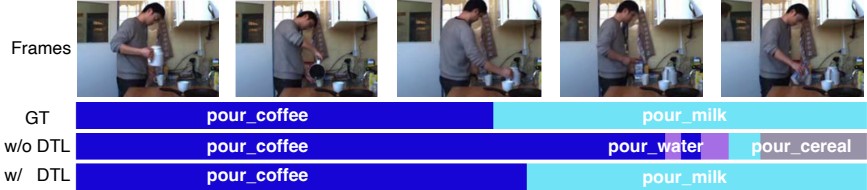

Figure 1: A video of activity "coffee preparation". The colored bars, from the top to the bottom, show the ground truth (GT), the predictions from a baseline [15], and the predictions from the baseline trained with DTL, respectively. Note that the baseline model erroneously predicts pour_cereal when DTL solves this problem with an exclusivity constraint between pour_coffee and pour_cereal.

## 1 Introduction

Recent years have witnessed significant advances in video action analysis tasks such as action recognition [29, 55], action detection [49], and temporal action segmentation [15, 1]. This can be credited to the availability of large-scale datasets [47, 23] and the development of effective deep visual backbones [4, 50, 17]. Although data-driven backbones can capture useful spatio-temporal features, learning a large number of highly diverse temporal dependencies and correlations over long time spans can be very challenging. In existing approaches, these temporal dependencies are not explicit to the model: it is possible to provide framewise annotation during training, however, temporal constraints like "event X has to occur after event Y" are still implicit in the annotations and not explicitly enforced. Even though these constraints could be statistically learned from a large amount of data, a pure vision model could still be confused by different actions with similar visual appearances. An

---

*This work was done when Ziwei Xu was visiting the Center for Research in Computer Vision.

36th Conference on Neural Information Processing Systems (NeurIPS 2022).

example of this is shown in Fig. 1, where the model confuses pour_cereal with pour_milk because both actions involve holding a carton.

In this paper, we propose a solution to this problem by employing temporal logic to apply declarative temporal constraints to the output of deep networks. Linear Temporal Logic (LTL) [43], for example, defines operators that describe necessities, possibilities, and dependencies of actions in a series of actions. Checking a network's output against those constraints (a.k.a. model checking) tells us if the predicted actions are logically correct. Logical correctness can be used as an additional training objective. According to prior works in the neuro-symbolic community [54, 52, 18], applying logic constraints improves the performance of deep networks in tasks for graph data [54] and images [52] by reducing logical errors in the output. Although one would anticipate similar benefits when using the temporal logic for action analysis, most of the focus in literature is still on non-sequential data.

In this work, we focus on the temporal aspect and consider temporal constraints in videos. Inspired by the foundational works in logic-based losses [54, 18], we propose a Differentiable Temporal Logic (DTL) framework, which uses an extended definition of linear temporal logic to constrain the output of action analysis models. At a high level, DTL treats the model output as a truth assignment of variables in LTL formulae. A differentiable evaluator performs model checking on the outputs and yields a satisfaction score, which measures the consistency between the constraints and the outputs. As we optimize the satisfaction score through standard optimization methods, the constraints are enforced on the deep network. Different from existing work [53], in DTL the relation between formula evaluation and formula satisfaction is deterministic, which means that DTL is logically sound. Moreover, we propose a comprehensive set of constraints covering both temporal and non-temporal dependencies between actions, and show how they are represented using DTL. We evaluate DTL on the challenging temporal action segmentation task, where modeling temporal dependencies between actions is crucial. The efficacy of our method is shown by the improved performance of different task models when constraints are enforced through DTL.

The contributions of this work are as follows:

- We present Differentiable Temporal Logic (DTL), a framework providing a model-agnostic manner of introducing temporal logic constraints to deep networks (cf. Section 3.2).

- We propose DTL constraints to describe a wide range of relations between actions, which can be automatically procured from dataset annotation (cf. Section 3.3).

- Experiments with different task models on the temporal action segmentation task show the efficacy of DTL. In addition, we provide an extensive study to show the effect of DTL constraints on task models.

## 2 Related Works

### 2.1 Temporal Action Segmentation

A temporal action segmentation model takes a set of video frames as input and predict the action category for each frame. The model needs to capture both short and long term dependencies between action categories. Earlier methods [3, 16] use sliding windows to model changes in visual appearance in a short time. Long-term dependencies are captured from those short-term information by sequential models like hidden Markov model [33] and recurrent networks [34, 45]. Despite the remarkable performance, when these models are used to process long videos they still face forgetting issues and heavy computation burden. Temporal Convolutional Network (TCN) [36, 37, 15, 38] enables efficient modelling of temporal dependency in variable time spans with its flexible receptive field. Many advancements are made based on MSTCN [15], which used stacked dilated temporal convolution layers for temporal modelling. For example, ASRF [28] uses a boundary regression task on top of MSTCN to improve the quality of segmentation. Huang *et al.* [26] proposed a graph-based dependency model to improve the modeling of relations between actions. Finally, Transformer [51] is introduced to this task and shows superior performance in [56]. Our view is that these methods can be complemented by DTL, since many declarative constraints are difficult to learn from the data. On the other hand, DTL can explicitly enforce these constraints on the task model during training.

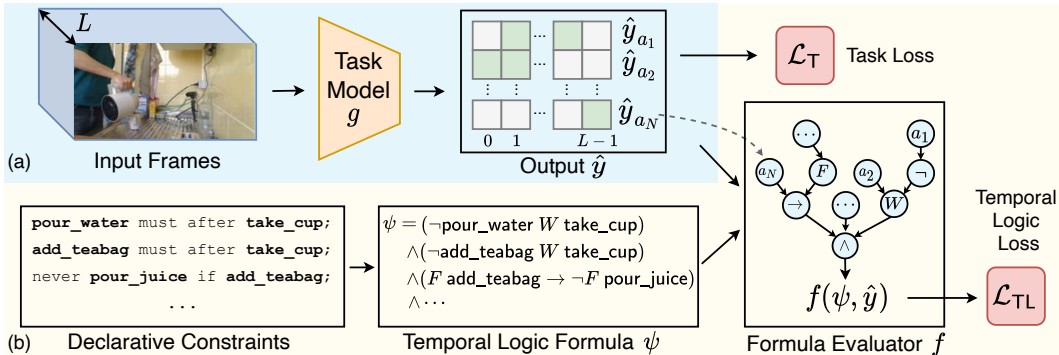

Figure 2: An overview of DTL framework. (a) shows a typical action segmentation pipeline, where a task model $g$ takes input frames and output segmentation $\hat{\boldsymbol{y}}_*$ for each action class. A task loss $\mathcal{L}_T$ is minimized during training. (b) shows the logic evaluation part of DTL. A set of declarative constraints on actions are converted into a temporal logic formula $\psi$. A formula evaluator $f$ takes $\psi$ and $\hat{\boldsymbol{y}}_*$ as input, and evaluates the consistency between the two, resulting in a logic loss $\mathcal{L}_{TL}$.

## 2.2 Temporal Logic and Action Analysis

Temporal logic is a large family of logic systems that specify the relations between timed events. Linear temporal logic (LTL) [43] is one of the earliest temporal logic. It is a propositional modal logic that states the necessity and possibility about events in a linear discrete succession of time steps. Later variants of LTL include Metric Temporal Logic (MTL) [40], which introduces time units as extra parameters to temporal operators, and Signal Temporal Logic [14], which specifies temporal properties for continuous signals. Allen's interval algebra [2] and Interval Temporal Logic (ITL) [22] specify the relations between intervals and their composition. Temporal logic has been used to specify constraints for program verification [43], robot control [13, 27, 44], and linguistics [41]. However, its application in action analysis remains scarce. CASE$^E$ [21] uses ITL as a structural language to describe atomic actions for event classification. T-LEAF [53] is the first (and so far the only, to the best of our knowledge) attempt to apply temporal logic to action analysis. It uses an embedding model to incorporate LTL formulae when training deep networks for action classification, which has little guarantee on logical soundness and only handles a small set of constraints.

DTL employs LTL for temporal constraints because LTL is: (1) more feasible than its successors with more operators and (2) expressive enough to benefit action analysis. In principle, DTL can be used with any logic system that is sound and whose formulae can be evaluated in a differentiable manner.

## 2.3 Logic Constraints on Neural Networks

Applying symbolic constraints to deep networks has been drawing increasing interest [25, 11, 39, 46, 20], yet it remains an open problem. An important challenge is that logic propositions generally take discrete truth values, while end-to-end differentiability is usually required to optimize a network. A workaround for this challenge is to find a continuous consistency measurement between network outputs and constraints. Existing methods can be categorized into two branches based on whether constraints are treated as data or procedures. The first branch is embedding-based [35, 30, 52, 53, 42], which treats constraints and network outputs as data and project them into a continuous space, where the distance between the two is used as a measurement. For example, LENSR [52] represents logical constraints in Deterministic Decomposable Negation Normal Form (d-DNNF) [8] and uses GCN [31] as the embedder. The second branch is procedure-based [54, 18, 19, 12, 24], which compiles constraints into computational procedures over network outputs and uses the result as consistency measurement. For example, semantic loss [54] compiles formulae into logic circuits [9], along which the probability of the output satisfying the formula is computed as the measurement. We design DTL as a procedure-based framework instead of an embedding-based one, since the latter generally lacks a soundness guarantee: high similarities in the embedding space does not guarantee logical correctness (or satisfaction).

# 3 Method

We consider the action segmentation task in a video clip with $L$ frames and $N$ candidate actions $\mathcal{A} = \{a_1, \ldots, a_N\}$. As shown in Fig. 2, we assume that there is a deep task model $g$ parameterized by $\boldsymbol{\Theta}$, which takes frames $\boldsymbol{x}_0, \ldots, \boldsymbol{x}_{L-1}$ as input, and outputs $\hat{\boldsymbol{y}}_{a_1}, \ldots, \hat{\boldsymbol{y}}_{a_N}$, where $\hat{\boldsymbol{y}}_{a_i} \in \mathbb{R}^L$ is the unnormalized score for the presence of action $a_i$ throughout the $L$ frames. We assume $\hat{\boldsymbol{y}}_{a_i,t} > 0$ indicates $a_i$ presents at time $t$, and $\hat{\boldsymbol{y}}_{a_i,t} \leq 0$ indicates the opposite. Apart from the ground truth label $\boldsymbol{y}_{a_1}, \ldots, \boldsymbol{y}_{a_N}$, there is a set of constraints that describe the relation between actions. The design goal of DTL is to incorporate these constraints into the task model $g$, so that $\hat{\boldsymbol{y}}$ complies with both the ground truth $\boldsymbol{y}$ and the constraints. In order to do so, we introduce a temporal logic representation $\Psi$, and an evaluator $f$ that enforces the constraints in formula $\psi \in \Psi$ on $g$, through its outputs $\hat{\boldsymbol{y}}$.

## 3.1 Syntax of Formulae

The definition of $\Psi$ is an extension of Linear Temporal Logic (LTL) [43]. A formula $\psi \in \Psi$ takes any of the forms separated by "|" below:

$$\psi := \mathsf{True} \mid \mathsf{False} \mid a \mid \neg\psi_1 \mid (\psi_1 \wedge \psi_2) \mid (\psi_1 \vee \psi_2) \mid \mathsf{X}\psi_1 \mid \mathsf{F}\psi_1 \mid (\psi_1 \mathsf{W} \psi_2) \mid (\psi_1 \mathsf{S} \psi_2), \quad (1)$$

where $a \in \mathcal{A}$ is the atomic proposition, and $\psi_1, \psi_2 \in \Psi$. The connectives X (NEXT), F (EVENTUAL), W (WEAK_UNTIL), and S (SINCE) are modal operators that form the temporal relations between propositions. In our context, atomic propositions $a$ represents action $a$. Note that the definition in Eqn. (1) is recursive. For example, if $a_1 \in \Psi$, then $\mathsf{X} \ldots \mathsf{XF}a_1 \in \Psi$. Semantically, $\mathsf{X}\psi$ is satisfied when proposition $\psi$ is satisfied in the next time step. $\mathsf{F}\psi$ is satisfied when $\psi$ is satisfied by the end of the sequence. $\psi_1 \mathsf{W} \psi_2$ being satisfied means that $\psi_1$ must always be satisfied until $\psi_2$ is satisfied (and $\psi_2$ might never be satisfied). $\psi_1 \mathsf{S} \psi_2$ being satisfied means $\psi_1$ is always satisfied after $\psi_2$ is satisfied. Section D.1 provides a more formal definition of these operators.

## 3.2 Formula Evaluator

A formula is said to be **satisfied** by a truth value assignment if the assignment is semantically compliant with the constraints. In the context of this paper, each atomic proposition $a_i$ is assigned $\hat{\boldsymbol{y}}_{a_i}$. Satisfiability is determined by evaluation, which is a function $f$ of a formula $\psi$ and $\hat{\boldsymbol{y}}$:

$$f_t(\psi, \hat{\boldsymbol{y}}) = f(\psi, \hat{\boldsymbol{y}}_{a_1:a_N,t:L-1}) : \Psi \times \underbrace{\mathbb{R}^{L-t} \times \cdots \times \mathbb{R}^{L-t}}_{N \text{ times}} \to \mathbb{R}, \quad (2)$$

where $\times$ refers to the Cartesian product. The two arguments of $f$ are formula $\psi$ and model output $\hat{\boldsymbol{y}}_{a_1:a_N}$. Parameter $t \in [0, L-1]$ is the start time of the evaluation. For example, $t = 2$ means that the evaluation is between $\psi$ and the prediction starting from the third frame, i.e. $\hat{\boldsymbol{y}}_{a_1:a_N,2:L-1}$. The result of $f$ is a satisfaction score that measures the consistency between $\hat{\boldsymbol{y}}$ and $\psi$.

Eqn. (2) is abstract and must be detailed for all possible forms of a formula $\psi$ can take in Eqn. (1). We aim to expand the definition so that $\Psi$ is logically sound, i.e. there is a determined relation between $f_t(\psi, \hat{\boldsymbol{y}})$ and the satisfaction of $\psi$ given $\hat{\boldsymbol{y}}$. Specifically, we would like $f_t(\psi, \hat{\boldsymbol{y}}) > 0$ to imply that $\psi$ is satisfied by $\hat{\boldsymbol{y}}$ at time $t$. Moreover, $\Psi$ must be differentiable to be incorporated with task models. Towards these goals, we first define the evaluation for constants and atomic propositions:

$$f_t(\mathsf{True}, \hat{\boldsymbol{y}}) = +\infty, \quad f_t(\mathsf{False}, \hat{\boldsymbol{y}}) = -\infty, \quad f_t(a, \hat{\boldsymbol{y}}) = \hat{\boldsymbol{y}}_{a,t}. \quad (3)$$

Indeed, the evaluation result for True and False will always be positive and negative, because the former is always satisfied and the latter is never satisfied. Note that if $\psi = a$, it is satisfied at time $t$ when $\hat{\boldsymbol{y}}_{a,t} > 0$, i.e. action $a$ happens at time $t$.

Next, we define the evaluation for operators "$\neg$" (negation), "$\wedge$" (logical and), and "$\vee$" (logical or):

$$f_t(\neg\psi_1, \hat{\boldsymbol{y}}) = -f_t(\psi_1, \hat{\boldsymbol{y}}), \quad (4)$$

$$f_t(\psi_1 \wedge \psi_2, \hat{\boldsymbol{y}}) = \min^\gamma \{f_t(\psi_1, \hat{\boldsymbol{y}}), f_t(\psi_2, \hat{\boldsymbol{y}})\}, \quad (5)$$

$$f_t(\psi_1 \vee \psi_2, \hat{\boldsymbol{y}}) = \max^\gamma \{f_t(\psi_1, \hat{\boldsymbol{y}}), f_t(\psi_2, \hat{\boldsymbol{y}})\}, \quad (6)$$

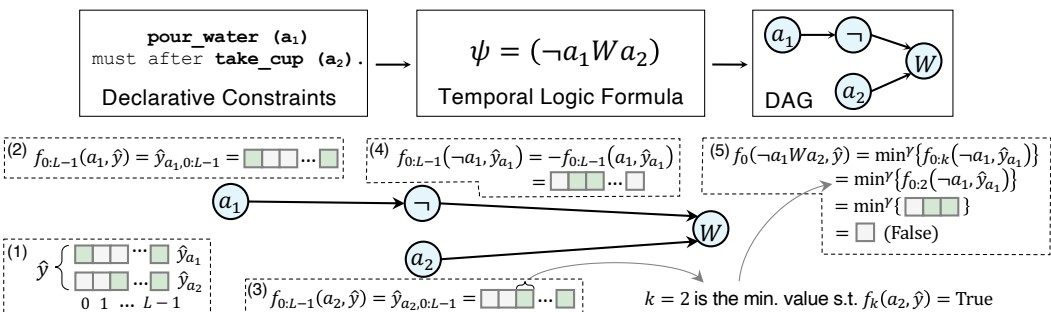

Figure 3: The evaluation of an example constraint on two actions over $L$ time steps. The constraint is first written as formula $\psi = (\neg a_1 \mathrm{W} a_2)$, then represented as a DAG. (1) – (5) show how $f_0(\neg a_1 \mathrm{W} a_2, \hat{\boldsymbol{y}})$ is computed. (1) shows the output $\hat{\boldsymbol{y}}$ of task model, where green boxes indicate positive values (True) and grey boxes indicate negative values (False). (2) and (3) show the evaluation of leaf nodes $a_1$ and $a_2$, which is the start of evaluation. (4) takes the evaluation results for $a_1$ from (2) and evaluate $\neg a_1$ following Eqn. (4). In (5), the W node uses the result for $\neg a_1$ from (4) and $a_2$ from (3) to perform the evaluation for $(\neg a_1 \mathrm{W} a_2)$ following Eqn. (9).

where $\gamma$ is a parameter of function $\min^\gamma\{x_{1:L-1}\} = -\frac{1}{\gamma} \ln \sum_{i=1}^{L-1} e^{-\gamma x_i}$, which approximates the minimum value of $\{x_{1:L-1}\}$ [6], and $\max^\gamma\{x_{1:L-1}\} = -\min^\gamma\{-x_{1:L-1}\}$. It can be shown [6] that $\lim_{\gamma \to \infty} \min^\gamma\{x_{1:L-1}\} = \min\{x_{1:L-1}\}$. In Eqn. (4), the operator $\neg$ flips the sign of $f_t(\psi_1, \hat{\boldsymbol{y}})$, reflecting the negation semantics of $\neg$. In Eqn. (5), $\min^\gamma\{f_t(\psi_1, \hat{\boldsymbol{y}}), f_t(\psi_2, \hat{\boldsymbol{y}})\}$ will be negative (False) if $\psi_1$ or $\psi_2$ or both are False, and will be positive (True) iff. both $\psi_1$ and $\psi_2$ are True. This is consistent with the semantics of $\wedge$. The same rationale applies to $\max^\gamma$ for $\vee$ in Eqn. (6).

Finally, we define the modal operators X, F, W, and S that are unique in $\Psi$. Intuitively, evaluating $\mathrm{X}\psi_1$ is equivalent to evaluating $\psi_1$ at the next time step. $\mathrm{F}\psi_1$ means there is at least one time step at which $\psi_1$ is satisfied. $\psi_1 \mathrm{W} \psi_2$ and $\psi_1 \mathrm{S} \psi_2$ require $\psi_1$ to be always satisfied in the time period specified by $\psi_2$. We formally define them as follows:

$$f_t(\mathrm{X}\psi_1, \hat{\boldsymbol{y}}) = f_{t+1}(\psi_1, \hat{\boldsymbol{y}}), \tag{7}$$

$$f_t(\mathrm{F}\psi_1, \hat{\boldsymbol{y}}) = \max^\gamma\{f_{t:L-1}(\psi_1, \hat{\boldsymbol{y}})\}, \tag{8}$$

$$f_t(\psi_1 \mathrm{W} \psi_2, \hat{\boldsymbol{y}}) = \min^\gamma\{f_{t:k}(\psi_1, \hat{\boldsymbol{y}})\}, \text{where } k \geq t \text{ is the min. integer s.t. } f_k(\psi_2, \hat{\boldsymbol{y}}) > 0, \tag{9}$$

$$f_t(\psi_1 \mathrm{S} \psi_2, \hat{\boldsymbol{y}}) = \min^\gamma\{f_{k:L-1}(\psi_1, \hat{\boldsymbol{y}})\}, \text{where } k \geq t \text{ is the min. integer s.t. } f_k(\psi_2, \hat{\boldsymbol{y}}) > 0. \tag{10}$$

In Eqn. (9), $\min^\gamma\{f_{t:k}(\psi_1, \hat{\boldsymbol{y}})\}$ is positive (True) iff. all elements of $\{f_{t:k}(\psi_1, \hat{\boldsymbol{y}})\}$ are positive, which means that $\psi_1$ stays True from time $t$ to time $k$. This is consistent with the semantics of W. We use $\min^\gamma$ for S in Eqn. (10) for the same reason.

The definitions in Eqn. (3)-(10) provide a soundness guarantee for $\Psi$, allowing us to use $f_t(\psi, \hat{\boldsymbol{y}}) > 0$ as an optimization objective to enforce the constraints in $\psi$ on a task model. Formally, when $\gamma \to \infty$, the approximated evaluation (because of $\min^\gamma$) becomes exact, and the following theorem is true by construction (a proof is provided in Section D.3):

**Theorem 1.** *(Soundness) With $\psi \in \Psi, \gamma \to \infty$: if $f_t(\psi, \hat{\boldsymbol{y}}) > 0$, then $\psi$ is satisfied by $\hat{\boldsymbol{y}}$ at time $t$.*

In essence, the evaluation process propagates network predictions from atomic propositions to logic operators and finally to the formula. Equivalently, we can represent a formula as a directed acyclic graph (DAG) where each leaf node is labeled with True, False, or action $a$; and each internal node is labeled with logic operators. The edges of the DAG point from child nodes to their parents, along which the truth value propagates following Eqn. (3)-(10). Fig. 3 illustrates how a constraint "a person cannot pour water before taking a cup" is converted to a DAG and evaluated.

## 3.3 Constraints

This section discusses different types of constraints we find useful for action analysis and the way they are represented using $\Psi$. A constraint can be categorized into either a temporal or a non-temporal constraint based on whether it states the possibility and necessity of an action in a specific

time period. We propose two temporal constraints, namely Backward Dependency (BD) and Forward Cancellation (FC), and two non-temporal constraints, namely Implication (Ip), and Exclusivity (Ex). We also introduce how constraints of these types are curated from the training annotations.

**Backward Dependency (BD)**   It is important to specify the proper order of actions because it usually determines the semantics of an activity. An order is generally related to temporal dependency: An action cannot be performed if its prerequisite actions did not occur. A way to describe this dependency is "one action must occur before another". We write this constraint as

$$\psi_{\text{BD}} = \wedge_{(a_i,a_j)\in\mathcal{B}}\text{BD}(a_i,a_j) = \wedge_{(a_i,a_j)\in\mathcal{B}}(\text{F}a_i \wedge \text{F}a_j) \rightarrow (\neg a_i \text{W} a_j),$$

where $(X \rightarrow Y)$ means "$X$ implies $Y$" and is equivalent to $(\neg X \vee Y)$. This logic expression means the following: if $a_i$ and $a_j$ occurs in the same video, $a_i$ must not occur until $a_j$ occurs. Set $\mathcal{B}$ contains action pairs that satisfy this constraint. In practice, $(a_1, a_2) \in \mathcal{B}$ if $a_2$ always occurs before $a_1$ for any co-occurrence of $a_1$ and $a_2$ in the annotation.

**Forward Cancellation (FC)**   Apart from backward dependence, actions can make some other actions impossible in all future time steps. For example, in a video of salad preparation, the action serve_salad_to_plate marks the end of the preparation. Once this action occurs, actions like cut_lettuce or cut_tomato should not occur thereafter. We write this constraint as

$$\psi_{\text{FC}} = \wedge_{(a_i,a_j)\in\mathcal{F}}\text{FC}(a_j,a_i) = \wedge_{(a_i,a_j)\in\mathcal{F}}(\text{F}a_i \wedge \text{F}a_j) \rightarrow (\neg a_i \text{S} a_j),$$

which reads "if $a_i$ and $a_j$ occur in the same video, $a_i$ cannot occur after the occurrence of $a_j$". Set $\mathcal{F}$ contains action pairs that satisfy this constraint.

**Implication (Ip)**   There could be some actions that are semantically dependent but temporally independent: these actions are necessary to complete an activity, but the order is not crucial. For example, in a video on juice preparation, the person must take_cup and peel_orange, but these two actions are not temporally correlated. We write this constraint as

$$\psi_{\text{Ip}} = \wedge_{(a_i,a_j)\in\mathcal{I}}\text{Ip}(a_i,a_j) = \wedge_{(a_i,a_j)\in\mathcal{I}}(\text{F}a_i \rightarrow \text{F}a_j).$$

Set $\mathcal{I}$ contains action pairs that satisfy this constraint. In practice, $(a_1, a_2) \in \mathcal{I}$ if $a_2$ always occurs if $a_1$ occurs in the annotation.

**Exclusivity (Ex)**   As opposed to what implication constraints describe, some actions are mutually exclusive: they will never co-occur in the same video. For example, if we know that a video contains a single activity that is "coffee" or "frying eggs", then pour_coffee cannot happen if put_egg_into_plate occurs at any time in the video. This constraint is written as

$$\psi_{\text{Ex}} = \wedge_{(a_i,a_j)\in\mathcal{X}}\text{Ex}(a_i,a_j) = \wedge_{(a_i,a_j)\in\mathcal{X}}(\text{F}a_i \rightarrow \neg\text{F}a_j).$$

Set $\mathcal{X}$ contains pairs of actions that never occur in the same video.

Finally, we connect all constraints using $\wedge$ as $\psi = \psi_{\text{BD}} \wedge \psi_{\text{FC}} \wedge \psi_{\text{Ip}} \wedge \psi_{\text{Ex}}$.

### 3.4   Training with Constraints

During training, we hope that the output $\hat{y}$ is constrained by both ground truth $y$ and logic constraints described by $\psi$. The constraints from ground truth are enforced by a task-specific task loss $\mathcal{L}_{\text{T}}(\hat{y}, y)$, for example, framewise cross-entropy loss for the temporal action segmentation task. For logic constraints, we treat $\hat{y}$ as an assignment of $\psi$ and from Theorem 1 we know $f_t(\psi, \hat{y}) > 0$ if $\hat{y}$ satisfies $\psi$ from time $t$. Therefore, for any $g$ we can minimize the following objective:

$$\mathcal{L} = \mathcal{L}_{\text{T}} + \lambda\mathcal{L}_{\text{TL}} = \mathcal{L}_{\text{T}}(\hat{y}, y) + \lambda\sigma\big(f_0(\psi, \hat{y})\big), \tag{11}$$

where $\mathcal{L}_{\text{T}}$ is the loss term for the target task, $\lambda$ is the weight of the loss of logic $\mathcal{L}_{\text{TL}}$, and $\sigma(x) = \log(1+e^{-x})$ penalizes negative evaluation results. We set $t = 0$ in $f_t(\psi, \hat{y})$ in our experiments, since we require the prediction to satisfy the constraints from the first frame. Note that it is possible to set $t$ to different values so that the constraints can be applied flexibly at different temporal locations.

Table 1: Results of action segmentation on 50Salads dataset.

| Task Model | | Edit | F1@10 | F1@25 | F1@50 | Acc |
|---|---|---|---|---|---|---|
| GRU | Base | $55.2 \pm 2.3$ | $63.7 \pm 2.3$ | $60.5 \pm 2.7$ | $53.4 \pm 2.9$ | $79.0 \pm 2.4$ |
| | Base + DTL | $62.1 \pm 1.6$ | $69.3 \pm 1.4$ | $66.5 \pm 1.8$ | $58.9 \pm 1.9$ | $80.3 \pm 2.2$ |
| | **Gain** | $\mathbf{7.1 \pm 2.8}$ | $\mathbf{5.6 \pm 2.7}$ | $\mathbf{6.0 \pm 3.2}$ | $\mathbf{5.5 \pm 3.6}$ | $\mathbf{1.3 \pm 3.3}$ |
| MS-TCN | Base [15] | 67.9 | 76.3 | 74.0 | 64.5 | 80.7 |
| | Base (Rerun) | $69.5 \pm 1.7$ | $75.7 \pm 1.7$ | $73.0 \pm 1.9$ | $64.5 \pm 2.2$ | $80.0 \pm 1.4$ |
| | Base + DTL | $70.5 \pm 1.0$ | $78.3 \pm 1.3$ | $76.5 \pm 1.1$ | $67.6 \pm 1.9$ | $81.5 \pm 1.5$ |
| | **Gain** | $\mathbf{1.0 \pm 2.0}$ | $\mathbf{2.6 \pm 2.1}$ | $\mathbf{3.5 \pm 2.2}$ | $\mathbf{3.0 \pm 2.9}$ | $\mathbf{1.5 \pm 2.0}$ |
| ASFormer | Base [56] | 79.6 | 85.1 | 83.4 | 76.0 | 85.6 |
| | Base (Rerun) | $76.9 \pm 0.9$ | $83.6 \pm 0.9$ | $81.5 \pm 0.8$ | $73.9 \pm 1.3$ | $84.2 \pm 1.2$ |
| | Base + DTL | $80.5 \pm 1.5$ | $87.1 \pm 1.3$ | $85.7 \pm 1.2$ | $78.5 \pm 1.6$ | $86.9 \pm 1.5$ |
| | **Gain** | $\mathbf{3.6 \pm 1.7}$ | $\mathbf{3.5 \pm 1.6}$ | $\mathbf{4.2 \pm 1.4}$ | $\mathbf{4.6 \pm 2.1}$ | $\mathbf{2.7 \pm 1.9}$ |

Table 2: Results of action segmentation on Breakfast dataset.

| Task Model | | Edit | F1@10 | F1@25 | F1@50 | Acc |
|---|---|---|---|---|---|---|
| GRU | Base | $56.8 \pm 2.0$ | $53.3 \pm 2.3$ | $48.4 \pm 2.5$ | $38.4 \pm 2.1$ | $70.0 \pm 1.7$ |
| | Base + DTL | $58.4 \pm 1.7$ | $56.5 \pm 1.1$ | $51.4 \pm 1.3$ | $40.7 \pm 2.1$ | $70.3 \pm 1.1$ |
| | **Gain** | $\mathbf{1.6 \pm 2.7}$ | $\mathbf{3.2 \pm 2.6}$ | $\mathbf{3.0 \pm 2.8}$ | $\mathbf{2.3 \pm 2.9}$ | $\mathbf{0.4 \pm 2.0}$ |
| MS-TCN | Base [15] | 61.7 | 52.6 | 48.1 | 37.9 | 66.3 |
| | Base (Rerun) | $71.2 \pm 1.4$ | $71.7 \pm 1.3$ | $65.7 \pm 1.5$ | $52.3 \pm 1.8$ | $71.3 \pm 1.2$ |
| | Base + DTL | $71.6 \pm 1.1$ | $73.0 \pm 0.4$ | $67.7 \pm 1.2$ | $54.4 \pm 0.8$ | $72.3 \pm 0.5$ |
| | **Gain** | $\mathbf{0.5 \pm 1.8}$ | $\mathbf{1.2 \pm 1.3}$ | $\mathbf{2.0 \pm 2.0}$ | $\mathbf{2.1 \pm 2.0}$ | $\mathbf{1.1 \pm 1.3}$ |
| ASFormer | Base [56] | 75.0 | 76.0 | 70.6 | 57.4 | 73.5 |
| | Base (Rerun) | $76.2 \pm 1.4$ | $77.8 \pm 1.2$ | $72.9 \pm 1.6$ | $60.5 \pm 1.7$ | $75.0 \pm 1.0$ |
| | Base + DTL | $77.7 \pm 1.6$ | $78.8 \pm 1.1$ | $74.5 \pm 1.5$ | $62.9 \pm 1.6$ | $75.8 \pm 0.9$ |
| | **Gain** | $\mathbf{1.5 \pm 2.1}$ | $\mathbf{1.1 \pm 1.6}$ | $\mathbf{1.6 \pm 2.2}$ | $\mathbf{2.4 \pm 2.4}$ | $\mathbf{0.8 \pm 1.4}$ |

# 4 Experiments

In this section, we first assess the proposed DTL using the temporal action segmentation task. Through this task, we show that DTL can provide dependency information to improve the performance of action analysis models, in both quantitative and qualitative manners. Then, we perform an ablation study to show the effects of different types of constraints. Finally, with a gradient-based analysis, we explain how DTL affects the task model. All experiments are run with PyTorch 1.10 on an NVIDIA A6000 GPU. More details are covered in the appendix.

## 4.1 Temporal Action Segmentation

**Datasets** We use 50Salads [48] and Breakfast [32] for this task. In these datasets, each video spans at least 200 seconds and contains at least five different actions. 50Salads contains 50 videos of salad preparation with frame-level action annotations of 19 actions. We generated a total of 313 constraints from its annotation. Breakfast consists of 1,712 videos with 18 video-level activities, and each frame has one of the 47 actions. We use only its action-level annotations for this task. There are a total of 2,145 constraints for this dataset. The details of the constraints are provided in Section E.

**Task Models** We use three task models in evaluation: a single-layer bidirectional Gated Recurrent Unit (GRU) [5], a temporal convolution model MSTCN [15], and a transformer model AS-Former [56]. We use all three task models to examine the performance gain brought by DTL. For the ablation study and further discussion, we use GRU and MSTCN for their ease of training. The architecture of GRU is detailed in the appendix. For experiments on MSTCN and ASFormer, we retrain the corresponding models using their released source codes. The task models are trained and assessed using the protocol in [15], where the inputs to the task models are the 2048-dimension

Table 3: Performance gain of GRU and MSTCN when trained with individual/all types of constraints. There are no exclusivity constraints in 50Salads.

(a) 50Salads

| Task Model | | Edit | F1@10 | F1@25 | F1@50 | Acc |
|---|---|---|---|---|---|---|
| GRU | +BD | 0.7 | -1.3 | -0.3 | -0.3 | 0.5 |
| | +FC | 7.0 | 5.5 | 6.6 | 6.3 | 2.1 |
| | +Ex | — | — | — | — | — |
| | +Ip | -0.8 | -0.9 | -0.8 | -1.2 | 0.6 |
| | +All | 7.1 | 5.6 | 6.0 | 5.5 | 1.3 |
| MSTCN | +BD | 0.1 | 0.3 | 0.6 | 0.0 | 0.1 |
| | +FC | 0.2 | 1.0 | 1.0 | 0.6 | 0.1 |
| | +Ex | — | — | — | — | — |
| | +Ip | 0.8 | 2.6 | 2.8 | 1.9 | 1.0 |
| | +All | 1.0 | 2.6 | 3.5 | 3.0 | 1.5 |

(b) Breakfast

| Task Model | | Edit | F1@10 | F1@25 | F1@50 | Acc |
|---|---|---|---|---|---|---|
| GRU | +BD | 0.4 | 0.8 | 0.3 | 0.2 | -0.2 |
| | +FC | 1.0 | 2.2 | 1.8 | 1.1 | -0.1 |
| | +Ex | 0.3 | 1.2 | 1.7 | 0.9 | 0.1 |
| | +Ip | 1.1 | 2.2 | 2.1 | 1.9 | 0.8 |
| | +All | 1.6 | 3.2 | 3.0 | 2.3 | 0.4 |
| MSTCN | +BD | 0.2 | 1.1 | 1.4 | 1.7 | 0.7 |
| | +FC | -0.1 | 0.8 | 1.3 | 0.8 | 0.2 |
| | +Ex | 0.1 | 0.7 | 0.8 | 1.1 | -0.7 |
| | +Ip | 0.0 | 0.4 | 1.0 | 1.0 | -0.2 |
| | +All | 0.5 | 1.2 | 2.0 | 2.1 | 1.1 |

features extracted using I3D [4] pre-trained on ImageNet [10]. Frame-wise cross-entropy is used as the task loss for all the task models. Levenshtein distance (Edit), F1 score with thresholds 0.1, 0.25, and 0.5 (F1@{10,25,50}), and frame-wise accuracy (Acc) are used to measure the quality of the outputs. The results are from a $k$-fold cross-validation, where $k = 5$ for 50Salads and $k = 4$ for Breakfast.

**Performance**  Table 1 and 2 show the performance of task models trained without (Base) and with (Base + DTL) temporal logic objective $\mathcal{L}_{\mathsf{TL}}$, and the performance difference (Gain), on 50Salads and Breakfast respectively. For completeness, we also include the base performance published in the original papers where applicable. A prominent observation is that the performance gain is consistently positive. This indicates the applicability and efficacy of DTL on task models with different architectures and on datasets of different scales. Another observation is that the gains on different metrics are not even: there are noticeable improvements on Levenshtein distance and F1 scores while the improvement on frame-wise accuracy is not as significant. We conjecture the reason to be that frame-wise accuracy treats frames independently and does not distinguish well between models with and without the over-segmentation problem. Since DTL constrains the relations between *segments* of actions instead of frames, metrics based on segment differences like Levenshtein distance and F1-scores are therefore comparably better reflections of performance improvement.

## 4.2  Ablation Study

To understand the effects of each constraint type, we perform an ablation study using GRU and MSTCN on both datasets. In each experiment, we apply only one type of constraint and retrain the task model. The results are shown in Table 3. The first observation is that while applying constraints helps the task model, the extent of improvement differs for different models on different datasets. On 50Salads, nearly all the action classes are present in each video sample, making Ip constraints less helpful for GRU. However, Ip is the most helpful constraint type for MSTCN, which indicates that MSTCN is comparably weaker in modelling action co-occurrence and benefits more from this constraint. On the other hand, the more complicated nature of Breakfast makes both models benefit from all four types of constraints. Another important observation is that more constraints do not guarantee better performance. For example, GRU enjoys the highest improvement with only FC constraints — higher than when BD and Ip are added. This reveals a trade-off between precision and completeness of constraints: Because the $\min^\gamma$ and $\max^\gamma$ functions used in Eqn. (5)-(10) are approximations, the evaluation becomes less accurate when the number of constraints increases. For MSTCN on 50Salads and both models on Breakfast, the benefit of more complete constraints outweighs the increased evaluation errors, resulting in more improvements when using all constraints. Understanding the fine-grained interactions between the constraints and different backbone models remains an open question.

## 4.3  Qualitative Results

We qualitatively compare predictions from task models trained with and without DTL in Fig. 4. In Fig. 4a, we show the output of MSTCN on 50Salads. Note that the baseline MSTCN

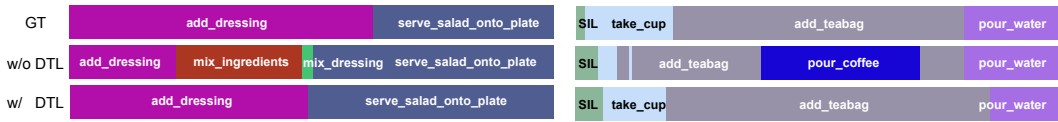

(a) Example output of MSTCN on 50Salads.  (b) Example output of GRU on Breakfast.

Figure 4: Qualitative comparison between model trained with and without DTL, which shows that logical errors are fixed by DTL. In each group, from the top to the bottom show the ground truth (GT), the prediction from baseline task model, and the prediction from task model trained with DTL.

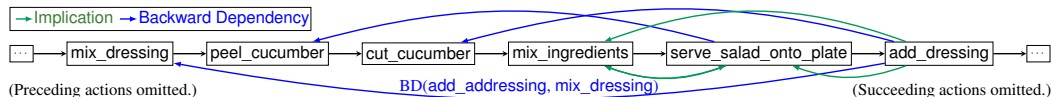

Figure 5: A snippet prediction from 50Salads and the most influential constraints for each segmentation. The black arrows represent temporal order. The colored arrows represent constraint types.

predicts a mix_dressing action after add_dressing, which is in conflict with a FC constraint FC(add_dressing, mix_dressing) because dressing cannot be mixed after it has been added to the salad. A similar example of GRU on Breakfast is shown in Fig. 4b, where the action pour_coffee is erroneously predicted in a video for "tea preparation". The baseline output violates an exclusivity constraint F add_teabag → ¬F pour_coffee because, except in special cases, coffee should not be added to tea. Both errors are fixed in the models trained with DTL. Moreover, we notice that correcting an individual wrong segment improves the quality of the whole output sequence. This is because the task models we consider are: (a) recurrent models that condition current output on past inputs, or (b) convolutional/transformer models that predict based on receptive field/attention information. In either case, correcting an error fixes the cascade effect it has on the model states, which improves the quality of all nearby outputs.

## 4.4 Analyzing the Effects of DTL

One of the greatest benefits of DTL is that it provides additional supervisory signals for temporal constraints, which are implicit in training data. We would like to understand how those signals affect the task model. Formally, given a model $g$ parameterized by $\Theta$ and an input $x$, we would like to know if a constraint $\psi_i$ promotes or suppresses the model's output about action $a$ at time $t$. We assume that the output of the model is continuous with respect to its parameters, or $\lim_{\|\delta\Theta\|\to 0} g_{\Theta+\delta\Theta}(x) = g_\Theta(x)$. With this assumption, the effect of $\psi_i$ on $\Theta$ can be approximated by the changes in the output as we update $\Theta$ based on $f_0(\psi_i, \hat{y})$. Specifically, we first compute $\delta^{\psi_i}\Theta = \partial\sigma(f_0(\psi_i, \hat{y}))/\partial\Theta$, where $\sigma(x) = \log(1 + e^{-x})$ as in Eqn. (11). Then, we update $\Theta$ as $\Theta' = \Theta - \gamma\delta^{\psi_i}\Theta$, where $\gamma = 10^{-4}$ is a small update step. Finally, the difference in output caused by $\psi_i$ is obtained as $\Delta^{\psi_i,x} = g_{\Theta'}(x) - g_\Theta(x)$, where $\Delta^{\psi_i,x} \in \mathbb{R}^{N \times L}$. A positive (negative) $\Delta^{\psi_i,x}_{a,t}$ indicates a promotive (suppressive) effect of $\psi_i$ on $\hat{y}_{a,t}$ as it increases (decreases) the score for action $a$ at time $t$.

We use $\Delta^{\psi_i,x}$ to pinpoint the constraint that makes a model predict an action $a$ at time $t$. This can be done by calculating $\Delta^{\psi_i,x}_{a,t}$ for all $\psi_i$ in $\psi$ and find the most positively influential constraint as $\arg\max_{\psi_i} \Delta^{\psi_i,x}_{a,t}$. For example, Fig. 5 shows a snippet of prediction from MSTCN on 50Salads and the constraints we found contributed the most to each predicted segment. One important observation from this is that actions in earlier time steps are generally promoted by later actions, as the former could serve as the prerequisites of the latter.

We can also summarize the total effect of $\psi_i$ on all time steps and samples as $\Delta^{\psi_i}_a = \frac{1}{|\mathcal{X}|} \sum_{x \in \mathcal{X}} \frac{1}{L} \sum_{t=0}^{L-1} \Delta^{\psi_i,x}_{a,t}$. $\Delta^{\psi_i}_a$ provides two pieces of information. The first is the effect of $\psi_i$ on all actions. As illustrated as a heatmap in Fig. 6, $\Delta^{\psi_i}_a$ suggest that the Ip constraints play mainly promotional roles, where the FC, BD, and Ex constraints are mostly suppressive. This is consistent with our design intuitions: Ip encourages co-occurrence of actions, FC and Ex constraints are intrinsically prohibitive, and BD constraints suppress an action if its prerequisites are missing.

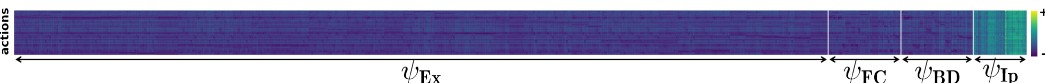

Figure 6: A heatmap showing $\mathbf{\Delta}_a^{\psi_i}$ of Breakfast, the vertical axis represents the 47 actions $a$ and the horizontal axis is the 2,145 constraints $\psi_i$. Labels at the bottom indicate constraint types. A brighter color indicates that a constraint has a more positive effect on an action.

Table 4: Actions in Breakfast dataset, and the most influential constraints. "$\sim$" refers to the action in the leftmost column of the row it is in.

| Action | Top Positive Constraint | Top Negative Constraint |
|---|---|---|
| stir_fruit | Ip(cut_fruit, $\sim$) | BD($\sim$, put_fruit2bowl) |
| stir_tea | Ip(add_teabag, $\sim$) | Ex(put_milk, $\sim$) |
| put_toppingOnTop | Ip(put_bunTogether, $\sim$) | FC(put_bunTogether, $\sim$) |

The second information is how an action $a$ is affected by other actions. This can be determined by selecting the most promotive and suppressive constraints for action $a$, and checking the actions involved in these constraints. Table 4 shows the relations between example actions in the Breakfast dataset. An interesting observation is that for put_ToppingOnTop the most promotive and suppressive action is the same put_PutBunTogether, meaning that relation between two actions could largely change depending on the context. This also indicates that DTL can help the model learn different dependencies between the same pair of actions.

## 5   Conclusion

We propose DTL, a framework that uses temporal logic to constrain the training of action analysis models. Experimental results on the action segmentation task show that DTL effectively improves the performance of task models with different architectures. An ablation study reveals the divergent effect of different types of rules on different task models. Our work suggests that temporal constraints can be explicitly provided to a deep network and reduce logical errors in its output. The source code of our work is accessible at `https://diff-tl.github.io/`.

**Limitations**   There remain some limitations in this work. First, in this work, we only explored a subset of temporal knowledge that is expressible as frequency matrices. When such temporal correlation is sparse, DTL is less effective. This calls for more flexible knowledge curation methods (e.g. with human involvement). Besides, non-temporal knowledge about actions, such as object affordance, can be exploited to handle more complicated actions like the verb-object compositions in Epic-Kitchens [7]. Second, in terms of knowledge type, DTL can benefit from more expressive logic languages, like Allen's Interval Algebra [2], which supports first-order temporal constraints beyond necessities and possibilities. Moreover, DTL formulae are evaluated in their raw forms, which means that the evaluation efficiency could be further improved by better formula compilation [9] and optimized message passing on directed acyclic graphs. In addition, the constraints we collected from dataset annotations only form a partial and clean view of real-world scenarios. The real-world constraints are more comprehensive and complex, which poses a higher requirement on generalization and robustness against uncertainties.

## Acknowledgments and Disclosure of Funding

This research is partially supported by the National Research Foundation, Singapore under its Strategic Capability Research Centres Funding Initiative. Any opinions, findings and conclusions or recommendations expressed in this material are those of the author(s) and do not reflect the views of National Research Foundation, Singapore.

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
