# Appendix for
# Don't Pour Cereal into Coffee: Differentiable Temporal Logic for Temporal Action Segmentation

**Ziwei Xu**[†][*]   **Yogesh S Rawat**[§]   **Yongkang Wong**[†]   **Mohan S Kankanhalli**[†]   **Mubarak Shah**[§]

† School of Computing, National University of Singapore
§ Center for Research in Computer Vision, University of Central Florida
{ziwei-xu,mohan}@comp.nus.edu.sg     yongkang.wong@nus.edu.sg
{yogesh,shah}@crcv.ucf.edu

## A   Table of Notations

The table below shows the notations grouped by the modules.

Table A1: Table of Notations

| | |
|---|---|
| **Model** | |
| $g_{\Theta}$ | A task model parameterized by $\Theta$ |
| $\boldsymbol{y}$ | Ground truth |
| $\hat{\boldsymbol{y}}$ | Output of the task model |
| $a_i$ | The $i^{\text{th}}$ action class, or its corresponding atomic proposition |
| $\psi$ | A DTL formula |
| $\Psi$ | The set of all DTL formulas |
| $f_t(\psi, \hat{\boldsymbol{y}})$ | Evaluation of $\hat{\boldsymbol{y}}$ against formula $\psi$ at time $t$ |
| **Sizes** | |
| $L$ | Length of input sequence |
| $N$ | Number of action classes |
| **Logic Operators** | |
| $\neg, \wedge, \vee$ | Logical NEGATION, logical AND, and logical OR |
| X | Temporal operator NEXT |
| F | Temporal operator EVENTUAL |
| W | Temporal operator WEAK_UNTIL |
| S | Temporal operator SINCE |
| **Constraints** | |
| $\text{BD}(a_i, a_j)$ | Action $a_i$ is backward dependent on action $a_j$ |
| $\text{FC}(a_i, a_j)$ | Action $a_i$ forward cancels action $a_j$ |
| $\text{Ip}(a_i, a_j)$ | Action $a_i$ implies action $a_j$ |
| $\text{Ex}(a_i, a_j)$ | Action $a_i$ excludes action $a_j$ |

---

[*]This work was done when Ziwei Xu was visiting the Center for Research in Computer Vision.

36th Conference on Neural Information Processing Systems (NeurIPS 2022).

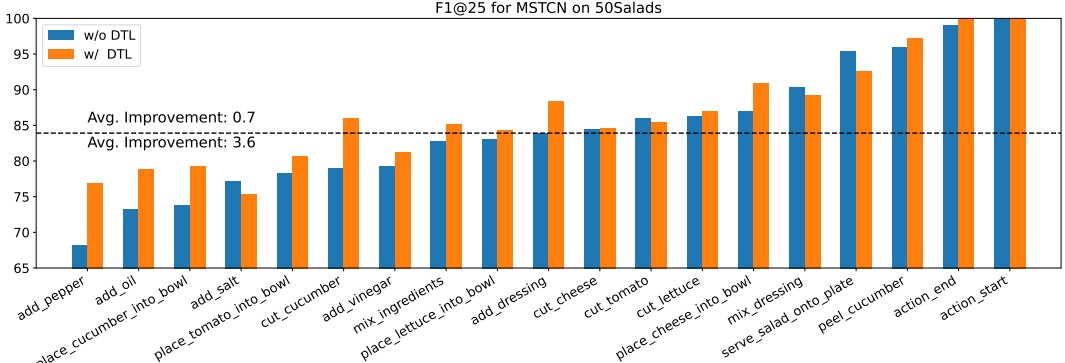

Figure A1: Class-wise F1@25 score for MSTCN on 50Salads. The classes on the horizontal axis are sorted based on the performance of the task model without DTL. Dashed line shows the median performance of all classes. The annotation above (below) the line indicates the averaged improvement for classes ranked at top (bottom) 50% in the baseline performance.

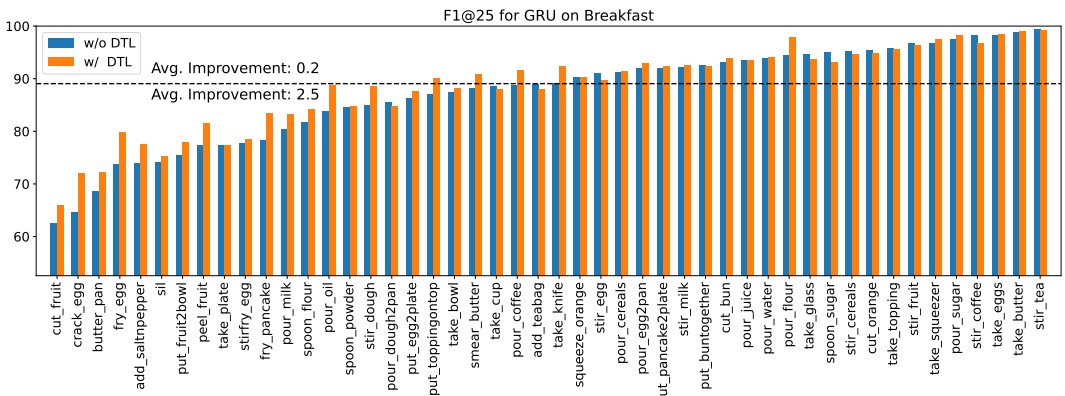

Figure A2: Class-wise F1@25 score for GRU on Breakfast. The classes on the horizontal axis are sorted based on the performance of the task model without DTL. Dashed line shows the median performance of all classes. The annotation above (below) the line indicates the average improvement for classes ranked at top (bottom) 50% in the baseline performance.

## B Implementation Details

**Task Models.** The implementation for MSTCN [2] and ASFormer [6] are from existing open-source code provided by corresponding authors. The GRU is implemented as follows: a fully-connected layer first transforms the 2048-dimension input into a 512-dimension vector. The vector is then sent to a bi-directional Gated Recurrent Unit layer with a hidden size of 512, which yields a 1024-dimension vector. The vector is finally transformed by a second fully-connected layer into an $N$-dimension vector, where each dimension represents the un-normalized score of an action class.

**Hyperparameters.** There are two hyperparameters in DTL: $\gamma$ and $\lambda$. $\gamma$ is the parameter for $\min^{\gamma}()$ and $\max^{\gamma}()$, and is set to 1. $\lambda$ is the weight for temporal logic loss. We set $\lambda = 0.1$ for both GRU and MSTCN, and $\lambda = 0.005$ for ASFormer. All hyperparameters are determined empirically.

**Training.** All the experiments are performed on an NVIDIA A6000 GPU with PyTorch 1.10. The training for MSTCN and ASFormer follows the way defined in the corresponding source code. We use Adam [3] to optimize the GRU with a learning rate of $5 \times 10^{-4}$ for 50 epochs. The model with the best validation performance is reported.

## C   Class-wise Performances

We provide class-wise performance (measured by F1@25) comparison between task models trained with and without DTL. The result is shown in Fig. A1 and Fig. A2. A general observation is that actions with lower performance on task models without DTL benefit more from DTL. We conjecture that actions with higher performance reflect the part of knowledge which is better learned by the task model from the annotation. Therefore, DTL is less effective for this subset of actions because currently its constraints are also procured from annotation. On the other hand, this shows that DTL is able to help the model better learn constraints from annotation. We anticipate more performance improvement with more general constraints that go beyond knowledge in the annotations in future works.

## D   Extended Discussion on DTL

Section 3.1 introduced logic operators and briefly introduced their semantics. In Section 3.2, we introduced the evaluation of those operators in detail. In this section, we extend the discussion on DTL and answer the following two questions:

1. What is the formal definition to the semantics of DTL operators (cf. Section D.1)?
2. What is the relation between the evaluation of DTL operators related and their semantics (cf. Section D.3)?

### D.1   Semantics of Logic Operators

This section formally defines the semantics of these logic operators, which is largely an extension of linear temporal logic. It is recommended to compare the semantics defined in Eqn. (A2)-(A9) below with the evaluation defined in Eqn. (3)-(10). Also recall that $\psi \in \Psi$ takes the following forms:

$$\psi \in \Psi := \mathsf{True} \mid \mathsf{False} \mid a \mid \neg\psi_1 \mid (\psi_1 \wedge \psi_2) \mid (\psi_1 \vee \psi_2) \mid \mathrm{X}\psi_1 \mid \mathrm{F}\psi_1 \mid (\psi_1 \mathrm{W}\psi_2) \mid (\psi_1 \mathrm{S}\psi_2). \quad \text{(A1)}$$

Before defining the semantics, we first introduce a symbol $\omega$, which is a truth assignment of $\psi$ in $L$ time steps. $\omega$ is an $L$-long word $\omega_{0:L-1} = \omega_0\omega_1 \ldots \omega_{L-1}$, where $\omega_t$ is the set of atomic propositions that are True at time $t^{\mathrm{b}}$. We use $\omega_{t_1:t_2}$ to denote a substring of $\omega$ from time $t_1$ to time $t_2$ and $\omega_{t_1:}$ as a shorthand for $\omega_{t_1:L-1}$. $\omega_{t:} \models \psi$ means $\omega_{t:}$ satisfies $\psi$ under the semantics of $\Psi$.

Now we start to define the semantics of $\Psi$. The semantics of constants are straight-forward:

$$\omega_{t:} \models \mathsf{True}, \quad \omega_{t:} \not\models \mathsf{False}, \quad \omega_{t:} \models a \text{ iff. } a \in \omega_t \quad \text{(A2)}$$

which states that any truth assignment satisfies $\psi$ if $\psi = \mathsf{True}$, and no truth assignment will satisfy $\psi$ if $\psi = \mathsf{False}$. If $\psi = a$, then a satisfying assignment must assign True to $a$ (make $a$ occur) at time 0.

For common propositional logic operators $\neg$ (NEGATION), $\wedge$ (AND) and $\vee$ (OR), we define their semantics as

$$\omega_{t:} \models \neg\psi_1 \qquad \text{iff. } \omega_{t:} \not\models \psi_1, \qquad\qquad \text{(A3)}$$

$$\omega_{t:} \models \psi_1 \wedge \psi_2 \qquad \text{iff. } \omega_{t:} \models \psi_1 \text{ and } \omega_{t:} \models \psi_2, \qquad \text{(A4)}$$

$$\omega_{t:} \models \psi_1 \vee \psi_2 \qquad \text{iff. } \omega_{t:} \models \psi_1 \text{ or } \omega_{t:} \models \psi_2, \qquad \text{(A5)}$$

where $\psi_1, \psi_2 \in \Psi$.

Finally, we define the semantics for modal operators $\mathrm{X}, \mathrm{F}, \mathrm{W}$, and $\mathrm{S}$:

$$\omega_{t:} \models \mathrm{X}\psi_1 \qquad \text{iff. } \omega_{t+1:} \models \psi_1, \qquad\qquad\qquad\qquad\qquad\qquad \text{(A6)}$$

$$\omega_{t:} \models \mathrm{F}\psi_1 \qquad \text{iff. } \exists t \le t_1 < L \text{ s.t. } \omega_{t_1:} \models \psi_1, \qquad\qquad\qquad \text{(A7)}$$

$$\omega_{t:} \models (\psi_1 \mathrm{W}\psi_2) \qquad \text{iff. } \exists t \le t_2 < L \text{ s.t. } \omega_{t_2:} \models \psi_2 \text{ and } \forall t \le t_1 < t_2, \omega_{t_1:} \models \psi_1, \qquad \text{(A8)}$$
$$\text{or } \forall t \le t_2 < L, \omega_{t_2:} \not\models \psi_2, \omega_{t:} \models \psi_1,$$

$$\omega_{t:} \models (\psi_1 \mathrm{S}\psi_2) \qquad \text{iff. } \exists t \le t_2 < L \text{ s.t. } \omega_{t_2:} \models \psi_2 \text{ and } \forall t_2 \le t_1 < L, \omega_{t_1:} \models \psi_1, \qquad \text{(A9)}$$
$$\text{or } \forall t \le t_2 < L, \omega_{t_2:} \not\models \psi_2.$$

Intuitively, X shifts the time to the next time step. F states the necessity of $\psi_1$. Both W and S state the possibility of $\psi_1$ during some time intervals specified by the occurrence of $\psi_2$.

---

[b]In the context of temporal action segmentation, $\omega_t$ is equivalent to the set of action that occurs at time $t$.

### D.2 Reviewing DTL Evaluation Rules

For easier reference, we copy Equation (3)-(10) from the main paper here. These equations defined the evaluation process for different operators:

$$f_t(\text{True}, \hat{\boldsymbol{y}}) = +\infty, \quad f_t(\text{False}, \hat{\boldsymbol{y}}) = -\infty, \quad f_t(a, \hat{\boldsymbol{y}}) = \hat{\boldsymbol{y}}_{a,t}. \tag{3}$$

$$f_t(\neg\psi_1, \hat{\boldsymbol{y}}) = -f_t(\psi_1, \hat{\boldsymbol{y}}), \tag{4}$$

$$f_t(\psi_1 \wedge \psi_2, \hat{\boldsymbol{y}}) = \min{}^\gamma\big\{f_t(\psi_1, \hat{\boldsymbol{y}}), f_t(\psi_2, \hat{\boldsymbol{y}})\big\}, \tag{5}$$

$$f_t(\psi_1 \vee \psi_2, \hat{\boldsymbol{y}}) = \max{}^\gamma\big\{f_t(\psi_1, \hat{\boldsymbol{y}}), f_t(\psi_2, \hat{\boldsymbol{y}})\big\}, \tag{6}$$

$$f_t(\text{X}\psi_1, \hat{\boldsymbol{y}}) = f_{t+1}(\psi_1, \hat{\boldsymbol{y}}), \tag{7}$$

$$f_t(\text{F}\psi_1, \hat{\boldsymbol{y}}) = \max{}^\gamma\big\{f_{t:L-1}(\psi_1, \hat{\boldsymbol{y}})\big\}, \tag{8}$$

$$f_t(\psi_1 \text{W}\psi_2, \hat{\boldsymbol{y}}) = \min{}^\gamma\{f_{t:k}(\psi_1, \hat{\boldsymbol{y}})\}, \text{where } k \geq t \text{ is the min. integer s.t. } f_k(\psi_2, \hat{\boldsymbol{y}}) > 0, \tag{9}$$

$$f_t(\psi_1 \text{S}\psi_2, \hat{\boldsymbol{y}}) = \min{}^\gamma\{f_{k:L-1}(\psi_1, \hat{\boldsymbol{y}})\}, \text{where } k \geq t \text{ is the min. integer s.t. } f_k(\psi_2, \hat{\boldsymbol{y}}) > 0. \tag{10}$$

### D.3 A Proof of Soundness for DTL

Theorem 1 in the main paper states the soundness of DTL: if $f_t(\psi, \hat{\boldsymbol{y}}) > 0$, then $\hat{\boldsymbol{y}}$ satisfies the constraints in formula $\psi$ at time $t$. In other words, it states that *evaluated satisfaction entails semantic satisfaction*. In this section, we provide a proof sketch for Theorem 1.

This proof starts with a basic assumption, where we formally establish the connection between $\hat{\boldsymbol{y}}$ and a truth assignment $\omega$:

**Assumption A1.** $\hat{\boldsymbol{y}}_{a_i,t} > 0 \Leftrightarrow a_i \in \omega_t$.

We also have the following assumption that rules out value "0" throughout the evaluation. This is necessary as $-0 = 0$ can break the evaluation of logical negation.

**Assumption A2.** $\hat{\boldsymbol{y}}_{a_i,t} \neq 0$, $\forall a \in \mathcal{A}$ *and* $0 \leq t < L$,

Then we can prove the following Theorem A1, which directly leads to Theorem 1.

**Theorem A1.** *With* $\psi \in \Psi, \gamma \to \infty$: $f_t(\psi, \hat{\boldsymbol{y}}) > 0 \Leftrightarrow \omega_{t:} \models \psi$.

*Proof.* Let $\psi, a, \psi_1, \psi_2 \in \Psi$, where $a$ is an atomic proposition. We need to prove Theorem 1 for all forms in Eqn. (A1). We do this by induction.

**Base Case** When $\psi = a$.

$$f_t(\psi, \hat{\boldsymbol{y}}) > 0 \xLeftrightarrow{\psi=a} f_t(a, \hat{\boldsymbol{y}}) > 0 \xLeftrightarrow{\text{Eqn. (3)}} \hat{\boldsymbol{y}}_{a,t} > 0 \xLeftrightarrow{\text{Asm. A1}} a \in \omega_t$$

$$\xLeftrightarrow{\text{Eqn. (A2)}} \omega_{t:} \models a \xLeftrightarrow{\psi=a} \omega_{t:} \models \psi.$$

Therefore the base case is true.

**Inductive Step** Inductive Hypothesis (I.H.): $f_t(\psi, \hat{\boldsymbol{y}}) > 0 \Leftrightarrow \omega_{t:} \models \psi$ for $\psi = \psi_1$ and $\psi = \psi_2$.

**Case 1:** $\psi = \neg\psi_1$.

- If $f_t(\psi, \hat{\boldsymbol{y}}) > 0$:

$$f_t(\psi, \hat{\boldsymbol{y}}) > 0 \xRightarrow{\psi=\neg\psi_1} f_t(\neg\psi_1, \hat{\boldsymbol{y}}) > 0 \xRightarrow{\text{Eqn. (4)}} -f_t(\psi_1, \hat{\boldsymbol{y}}) > 0$$

$$\Rightarrow f_t(\psi_1, \hat{\boldsymbol{y}}) < 0 \xRightarrow{\text{I.H.}} \omega_{t:} \not\models \psi_1 \xRightarrow{\text{Eqn. (A3)}} \omega_{t:} \models \neg\psi_1$$

$$\xRightarrow{\psi=\neg\psi_1} \omega_{t:} \models \psi.$$

- If $\omega_{t:} \models \psi$:

$$\omega_{t:} \models \psi \xrightarrow{\psi=\neg\psi_1} \omega_{t:} \models \neg\psi_1 \xrightarrow{\text{Eqn. (A3)}} \omega_{t:} \not\models \psi_1 \xrightarrow{\text{I.H. and Asm. A2}} f_t(\psi_1, \hat{\boldsymbol{y}}) < 0$$

$$\Rightarrow -f_t(\psi_1, \hat{\boldsymbol{y}}) > 0 \xrightarrow{\text{Eqn. (4)}} f_t(\neg\psi_1, \hat{\boldsymbol{y}}) > 0$$

$$\xrightarrow{\psi=\neg\psi_1} f_t(\psi, \hat{\boldsymbol{y}}) > 0.$$

**Case 2:** $\psi = (\psi_1 \wedge \psi_2)$.

$$f_t(\psi, \hat{\boldsymbol{y}}) > 0 \xleftrightarrow{\psi=(\psi_1\wedge\psi_2)} f_t(\psi_1 \wedge \psi_2, \hat{\boldsymbol{y}}) > 0 \xleftrightarrow{\text{Eqn. (5)}} \min\{f_t(\psi_1, \hat{\boldsymbol{y}}), f_t(\psi_2, \hat{\boldsymbol{y}})\} > 0$$

$$\Leftrightarrow f_t(\psi_1, \hat{\boldsymbol{y}}) > 0 \text{ and } f_t(\psi_2, \hat{\boldsymbol{y}}) > 0 \xleftrightarrow{\text{I.H.}} \omega_{t:} \models \psi_1 \text{ and } \omega_{t:} \models \psi_2$$

$$\xleftrightarrow{\text{Eqn. (A4)}} \omega_{t:} \models (\psi_1 \wedge \psi_2) \xleftrightarrow{\psi=(\psi_1\wedge\psi_2)} \omega_{t:} \models \psi.$$

**Case 3:** $\psi = (\psi_1 \vee \psi_2)$. The proof is similar to Case 2.

**Case 4:** $\psi = \mathbf{X}\psi_1$.

$$f_t(\psi, \hat{\boldsymbol{y}}) > 0 \xleftrightarrow{\psi=\mathbf{X}\psi_1} f_t(\mathbf{X}\psi_1, \hat{\boldsymbol{y}}) > 0 \xleftrightarrow{\text{Eqn. (7)}} f_{t+1}(\psi_1, \hat{\boldsymbol{y}}) > 0$$

$$\xleftrightarrow{\text{I.H.}} \omega_{t+1:} \models \psi_1 \xleftrightarrow{\text{Eqn. (A6)}} \omega_{t:} \models \mathbf{X}\psi_1 \xleftrightarrow{\psi=\mathbf{X}\psi_1} \omega_{t:} \models \psi.$$

**Case 5:** $\psi = \mathbf{F}\psi_1$.

$$f_t(\psi, \hat{\boldsymbol{y}}) > 0 \xleftrightarrow{\psi=\mathbf{F}\psi_1} f_t(\mathbf{F}\psi_1, \hat{\boldsymbol{y}}) > 0 \xleftrightarrow{\text{Eqn. (8)}} \max\{f_{t:L-1}(\psi_1, \hat{\boldsymbol{y}})\} > 0$$

$$\Leftrightarrow \exists t \le t_1 < L \text{ s.t. } f_{t_1}(\psi_1, \hat{\boldsymbol{y}}) > 0 \xleftrightarrow{\text{I.H.}} \omega_{t_1:} \models \psi_1$$

$$\xleftrightarrow{\text{Eqn. (A7)}} \omega_{t:} \models \mathbf{F}\psi_1 \xleftrightarrow{\psi=\mathbf{F}\psi_1} \omega_{t:} \models \psi.$$

**Case 6:** $\psi = (\psi_1\mathbf{W}\psi_2)$.

- If $\exists k \in [t, L)$ s.t. $f_k(\psi_2, \hat{\boldsymbol{y}}) > 0$:

$$f_t(\psi, \hat{\boldsymbol{y}}) > 0 \xleftrightarrow{\psi=(\psi_1\mathbf{W}\psi_2)} f_t(\psi_1\mathbf{W}\psi_2, \hat{\boldsymbol{y}}) > 0$$

$$\xleftrightarrow{\text{Eqn. (9)}} \min\{f_{k':t}(\psi_1, \hat{\boldsymbol{y}}) > 0\}, \text{ and } k' \in [t, L) \text{ is the min. integer s.t. } f_{k'}(\psi_2, \hat{\boldsymbol{y}}) > 0$$

$$\Leftrightarrow \exists t_1 \in [t, k] \text{ s.t. } f_{t_1}(\psi_1, \hat{\boldsymbol{y}}) > 0 \text{ and } \exists k \in [t, L) \text{ s.t. } f_k(\psi_2, \hat{\boldsymbol{y}}) > 0$$

$$\xleftrightarrow{\text{I.H.}} \exists t_1 \in [t, k] \text{ s.t. } \omega_{t_1:} \models \psi_1 \text{ and } \exists k \in [t, L) \text{ s.t. } \omega_{k:} \models \psi_2$$

$$\xleftrightarrow{\text{Eqn. (A8)}} \omega_{t:} \models (\psi_1\mathbf{W}\psi_2) \xleftrightarrow{\psi=(\psi_1\mathbf{W}\psi_2)} \omega_{t:} \models \psi.$$

- If $\forall k \in [t, L), f_k(\psi_2, \hat{\boldsymbol{y}}) < 0$: This is a special case for W (and S as well). When this happens, we directly set $f_t(\psi, \hat{\boldsymbol{y}}) = \min^\gamma(f_{t:L-1}(\psi_1, \hat{\boldsymbol{y}}))$. Then $f_t(\psi, \hat{\boldsymbol{y}}) > 0 \Leftrightarrow \omega_{t:} \models \psi_1 \Leftrightarrow \omega_{t:} \models (\psi_1\mathbf{W}\psi_2)$.

**Case 7:** $\psi = (\psi_1\mathbf{S}\psi_2)$. The proof is similar to Case 6. $\qquad\square$

# E Constraints

This section provides a quick view of the constraints used in our experiment. We first explain how constraints are collected and then provide samples for different types of constraints collected for the two datasets used in our experiment.

### E.1 Collecting Constraints

The constraints are automatically generated from the existing annotations of datasets. Algorithm A1 shows how statistics about co-occurrences between actions can be collected. Then Algorithm A2 uses those statistics to generate the four types of constraints discussed in Section 3.3.

---

**Algorithm A1:** Algorithm to collect the co-occurrence statistics from dataset annotation.

---

**Input:** Set of samples $\mathcal{M} = \{\boldsymbol{m}_1, \boldsymbol{m}_2, \ldots, \boldsymbol{m}_M\}$, where $\boldsymbol{m}_i = [y_0, y_2, ..., y_{L-1}]$ and $y_i \in \mathcal{A}$
  is the index of one of the $N$ actions.

▷ $\boldsymbol{B}[a_i, a_j]$ is the frequency of $a_i$ occurring before $a_j$
▷ $\boldsymbol{P}[a_i, a_j]$ is the frequency of $a_i$ occurring after $a_j$
▷ $\boldsymbol{J}[a_i, a_j]$ is the number of videos where $a_i$ occurs with $a_j$
▷ $\boldsymbol{C}[a_i, a_j]$ is the number of videos where $a_i$ occurs

1  $\boldsymbol{B}, \boldsymbol{P}, \boldsymbol{J} \leftarrow$ zero matrices of size $N \times N$;
2  $\boldsymbol{C} \leftarrow$ zero vector of size $N$;
3  **foreach** $m \in \mathcal{M}$ **do**
4      occur_flags $\leftarrow$ zero vector of size $N$;
5      co_occur_flags $\leftarrow$ zero matrix of size $N \times N$;
6      $y_0, y_1, ..., y_{L-1} \leftarrow$ annotation of $m$;
7      **foreach** $t \in 0, 1, \ldots, L - 1$ **do**
8          **if** *occur_flags[$y_t$] == 0* **then**
9              $\boldsymbol{C}[y_t] \leftarrow \boldsymbol{C}[y_t] + 1$ ;
10             occur_flags[$y_t$] $\leftarrow 1$;
11         **foreach** $u \in \{0, 1, \ldots, t\}$ **do**
12             $\boldsymbol{B}[y_u, y_t] \leftarrow \boldsymbol{B}[y_u, y_t] + 1$;
13             **if** *co_occur_flags[$y_t, y_u$] == 0* **then**
14                 $\boldsymbol{J}[y_u, y_t] \leftarrow \boldsymbol{J}[y_u, y_t] + 1$ ;
15                 co_occur_flags[$y_u, y_t$] = 1;
16         **foreach** $u \in \{t + 1, t + 2, \ldots, L - 1\}$ **do**
17             $\boldsymbol{P}[y_u, y_t] \leftarrow \boldsymbol{P}[y_u, y_t] + 1$;
18             **if** *co_occur_flags[$y_t, y_u$] == 0* **then**
19                 $\boldsymbol{J}[y_u, y_t] \leftarrow \boldsymbol{J}[y_u, y_t] + 1$ ;
20                 co_occur_flags[$y_u, y_t$] = 1;

21 **return** $\{\boldsymbol{B}, \boldsymbol{P}, \boldsymbol{J}, \boldsymbol{C}\}$;

---

### E.2 Samples of Constraints

For clarity, we show 10 entries for each type of constraint.

#### E.2.1 Breakfast

Breakfast contains 48 actions for ten different activities about breakfast preparation. Each video contains a single activity, which could be making coffee, cereal, tea, fried egg, pancake, sandwich, juice, etc. Breakfast is therefore different from 50Salads because its actions could be mutually exclusive. Below is a list of actions:

- take_cup
- pour_coffee
- pour_milk
- pour_sugar
- stir_coffee
- spoon_sugar
- add_teabag

- pour_water
- stir_tea
- cut_bun
- smear_butter
- put_toppingOnTop
- put_bunTogether
- take_plate

- take_knife
- take_butter
- take_topping
- cut_orange
- squeeze_orange
- take_glass
- pour_juice

**Algorithm A2:** Algorithm to generate the constraints from the collected statistics.

---

**Input:** Statistics $\{B, P, J, C\}$ collected by Algorithm A1.

▷ $B[a_i, a_j]$ is the frequency of $a_i$ occurring before $a_j$
▷ $P[a_i, a_j]$ is the frequency of $a_i$ occurring after $a_j$
▷ $J[a_i, a_j]$ is the number of videos where $a_i$ occurs with $a_j$
▷ $C[a_i, a_j]$ is the number of videos where $a_i$ occurs

1 Constraints ← {} ;
▷ Empty set of rules
2 **foreach** $i \in \{0, 1, \ldots, N-1\}$ **do**
3    **foreach** $j \in \{0, 1, \ldots, N-1\}$ **do**
4       **if** $i \neq j$ *and* $B[i, j] == 0$ **then**
         ▷ action i is "backward dependent" on j
5          Constraints ← append_BD$(i, j)$ ;
6       **if** $J[i, j] > 0$ *and* $P[i, j] == 0$ **then**
         ▷ action j "forward cancels" j
7          Constraints ← append_FC$(j, i)$ ;
8       **if** $i \neq j$ *and* $J[i, j]/C[j] == 1$ **then**
         ▷ action j implies i
9          Constraints ← append_Ip$(j, i)$ ;
10       **if** $i \neq j$ *and* $J[i, j] == 0$ **then**
         ▷ action i and j is exclusive
11          Constraints ← append_Ex$(i, j)$ ;

12 **return** Constraints;

---

- take_squeezer
- take_bowl
- pour_cereals
- stir_cereals
- spoon_powder
- stir_milk
- pour_oil
- take_eggs
- crack_egg
- add_saltnpepper

- fry_egg
- put_egg2plate
- butter_pan
- cut_fruit
- put_fruit2bowl
- peel_fruit
- stir_fruit
- stirfry_egg
- stir_egg
- pour_egg2pan

- spoon_flour
- stir_dough
- pour_dough2pan
- fry_pancake
- put_pancake2plate
- pour_flour
- SIL

**Backward Dependency** The following shows a subset of the back dependency constraints, where $BD(a_i, a_j) = (Fa_i \wedge Fa_j) \rightarrow (\neg a_i W a_j)$ reads "action $a_i$ is backward dependent on action $a_j$."

- BD(pour_dough2pan, spoon_flour),
- BD(squeeze_orange, cut_orange),
- BD(add_saltnpepper, take_bowl),
- BD(put_egg2plate, pour_egg2pan),
- BD(stir_coffee, pour_coffee),

- BD(butter_pan, take_eggs),
- BD(pour_dough2pan, butter_pan),
- BD(stir_tea, take_cup),
- BD(pour_milk, pour_coffee),
- BD(pour_juice, take_plate).

**Forward Cancellation** The following shows a subset of the forward cancellation constraints, where $FC(a_i, a_j) = (Fa_i \wedge Fa_j) \rightarrow (\neg a_j S a_i)$ reads "action $a_i$ cancels the future occurrence of $a_j$."

- FC(pour_dough2pan, pour_flour),
- FC(take_squeezer, take_knife),
- FC(fry_pancake, pour_dough2pan),
- FC(put_buntogether, take_topping),
- FC(pour_dough2pan, take_bowl),
- FC(put_toppingontop, cut_bun),
- FC(stir_fruit, peel_fruit),
- FC(pour_milk, pour_coffee),
- FC(put_buntogether, take_knife),
- FC(stir_coffee, take_cup).

**Implication** The following shows a subset of the implication constraints, where $Ip(a_i, a_j) = Fa_i \rightarrow Fa_j$ reads "action $a_i$ implies the occurrence of $a_j$."

- Ip(fry_pancake, pour_dough2pan),
- Ip(pour_flour, stir_dough),
- Ip(butter_pan, crack_egg),
- Ip(stir_dough, pour_milk),
- Ip(pour_cereals, pour_milk),
- Ip(add_teabag, pour_water),
- Ip(spoon_powder, pour_milk),
- Ip(take_topping, cut_bun),
- Ip(take_butter, smear_butter),
- Ip(stirfry_egg, crack_egg).

**Exclusivity** The following shows a subset of the exclusivity constraints, where $Ex(a_i, a_j) = Fa_i \rightarrow \neg Fa_j$ reads "if action $a_i$ occurs, action $a_j$ will not occur in the same video".

- Ex(butter_pan, cut_bun),
- Ex(smear_butter, spoon_sugar),
- Ex(take_butter, pour_sugar),
- Ex(stir_cereals, cut_bun),
- Ex(add_teabag, take_butter),
- Ex(take_butter, stir_cereals),
- Ex(put_fruit2bowl, put_pancake2plate),
- Ex(put_fruit2bowl, pour_sugar),
- Ex(cut_bun, spoon_sugar),
- Ex(spoon_flour, add_teabag).

### E.2.2 50Salads

50Salads contains 19 actions for a single activity "making salad":

- cut_tomato
- place_tomato_into_bowl
- cut_cheese
- place_cheese_into_bowl
- cut_lettuce
- place_lettuce_into_bowl
- add_salt
- add_vinegar
- add_oil
- add_pepper
- mix_dressing
- peel_cucumber
- cut_cucumber
- place_cucumber_into_bowl
- add_dressing
- mix_ingredients
- serve_salad_onto_plate
- action_start
- action_end

**Backward Dependency**

- BD(serve_salad_onto_plate, peel_cucumber),
- BD(serve_salad_onto_plate, place_tomato_into_bowl),
- BD(add_dressing, cut_tomato),
- BD(serve_salad_onto_plate, cut_cucumber),
- BD(serve_salad_onto_plate, cut_lettuce),
- BD(add_dressing, add_pepper),
- BD(serve_salad_onto_plate, mix_ingredients),
- BD(serve_salad_onto_plate, add_salt),
- BD(serve_salad_onto_plate, place_cucumber_into_bowl),
- BD(add_dressing, cut_cheese).

**Forward Cancellation**

- FC(add_dressing, peel_cucumber),
- FC(serve_salad_onto_plate, add_vinegar),
- FC(place_cheese_into_bowl, cut_cheese),
- FC(mix_ingredients, cut_cheese),
- FC(add_dressing, add_oil),

- FC(place_cucumber_into_bowl, peel_cucumber),
- FC(serve_salad_onto_plate, mix_ingredients),
- FC(serve_salad_onto_plate, cut_cheese),
- FC(serve_salad_onto_plate, peel_cucumber),
- FC(add_dressing, place_cucumber_into_bowl).

**Implication**

- Ip(place_cheese_into_bowl, action_end),
- Ip(cut_cheese, cut_lettuce),
- Ip(add_oil, place_tomato_into_bowl),
- Ip(place_tomato_into_bowl, cut_tomato),
- Ip(mix_dressing, cut_cheese),

- Ip(add_salt, add_pepper),
- Ip(place_lettuce_into_bowl, add_oil),
- Ip(add_salt, place_tomato_into_bowl),
- Ip(place_tomato_into_bowl, action_end),
- Ip(add_oil, cut_tomato).

**Exclusivity**  There are no exclusivity constraints because all actions are from the same activity.

## F   Action Detection on Charades

One might be curious about whether DTL works without the "one-action-per-frame" assumption used in the main paper. In fact, this assumption emerges from the definition of the action segmentation task rather than DTL itself.

To examine the applicability of DTL without this assumption, we assess DTL using the action detection task on the Charades [5] dataset. Different from the datasets used in the action segmentation task, every frame in Charades can be labeled with zero or multiple actions. The dataset contains 9,848 videos with 157 frame-level action categories. Each video spans about 30 seconds and contains six actions on average. Following the same procedure introduced in Section 3.3, we collected a total of 9,668 constraints. We sampled 2,000 constraints for this experiment.

Table A2: Performances on the Charades dataset.

| Task Model | | mAP (%) |
| --- | --- | --- |
| GRU | Base | 20.7 |
| | Base + DTL | 21.6 |
| | **Gain** | **0.9** |
| TConv | Base | 17.2 |
| | Base + DTL | 18.3 |
| | **Gain** | **1.1** |

Following the procedures in [4], we use I3D [1] pretrained on Kinetics-400 to extract the frame-level features and mean average precision (mAP) as the performance metric. DTL is assessed using two task models, namely, a GRU with 512 hidden units (abbreviated as GRU), and a three-layer temporal convolutional network (abbreviated as TConv). As shown in Table A2, improvements are observed for both task models when they are trained with DTL. Note that since we aim to show that DTL works across different tasks, we do not use constraints unique to the action detection task. For example, we do not collect constraints about co-occurrences of actions in the same frame. This means further improvement is possible when those absent constraints are collected and enforced. The definition of a more comprehensive set of constraints and their applications in various action analysis tasks are left as future works.