# OpenReview forum: "Don't Pour Cereal into Coffee: Differentiable Temporal Logic for Temporal Action Segmentation"
_NeurIPS.cc/2022/Conference — NeurIPS 2022 Accept_

### Official Review · Reviewer_TJYF · 2022-07-08

**Rating:** 5
**Confidence:** 5
**Soundness:** 3 good
**Presentation:** 3 good
**Contribution:** 3 good

**Summary:**

The paper introduces a Differentiable Temporal Logic (DTL), a framework providing a model-agnostic manner of introducing temporal logic constraints to deep networks which can describe relations between actions.


**Questions:**

To conclude the paper needs more discussions/experiments to prove its claims.

- First, it needs a more elaborated comparison with other methods following the same reasoning, such as [1, 2].

- Secondly, it needs experiments on more complex datasets such as Epic-Kitchen or Charades, to better understand the impact of the proposed approach.


After rebuttal, the authors answered all my questions convincingly and provided essential details about their proposed method.

**Limitations:**

yes

**Strengths And Weaknesses:**

Strengths :
    Enable to define a priori-knowledge (i.e. temporal logic constraints) to improve the reasoning on events (i.e. temporal relations).

Weaknesses:

- The contribution is limited:
1) First of all the authors should compare their approach to other methods [1, 2] that use this kind of reasoning on events (i.e. temporal relations) to improve the predictions. For example, unlike the proposed approach, [1] does not use a priori-knowledge but they learn the relations between the events. Hence, it could be a good practice to compare the two approaches, as learnt constraints are often more convenient than manually defined constraints.
2) Also it could be interesting to combine both approaches to see if a priori-knowledge and learnt logical constraints can bring improvement to work that learns events relations directly from the data. How complementary are both approaches?
3) Moreover, it is not clear whether the model can work in case of datasets where different actions can happen at the same time (co-occurrent events) as in Epic-Kitchen?
4) The logical constraints may be new to computer vision but it has been used for a long time on deep learning and since the contribution they provide is not on computer vision (add a logical loss that was used in other deep learning studies) makes the contribution limited.

- The method is very data-specific: experiment on more datasets is needed.
 5) The proposed Declarative Constraints Temporal/Logic Formula is easy to use in datasets such as breakfast and 50 salades (datasets with fixed scripts). However in the case of Epic-Kitchen, there are more events and more complex actions, where it is more challenging to define event constraints and so results on this dataset can better prove the authors claims.
6) The proposed model shows improvement especially in the case of small amounts of training data and annotations as their results show a bigger improvement on 50Salades compared to breakfast.
7) The results show 2 important observations: Firstly, when the base network can better model the temporal relations, the improvement brought by the proposed approach is less important as it is the case of MSTCN and Asformer compared to GRU. Secondly, when the dataset is bigger (more instances), the improvement is not as important, as we can see from comparing improvement on 50Salades and BreakFast.
So, the impact is limited.

[1] Huang, Y., Sugano, Y., & Sato, Y. (2020). Improving action segmentation via graph-based temporal reasoning. In Proceedings of the IEEE/CVF conference on computer vision and pattern recognition (pp. 14024-14034).
[2] R. Dai, S. Das and F. Bremond. CTRN: Class Temporal Relational Network For Action Detection. In Proceedings of the 32nd British Machine Vision Conference, BMVC 2021, United Kingdom, Virtual, November 22-25, 2021.

---

> ### Author Response · Authors · 2022-08-02
> **Thank you very much for the comments. In summary, below are our responses to the raised issues regarding comparison with other temporal reasoning methods, contributions, datasets, and evaluations.**
>
> > The authors should compare their approach to other methods [1, 2] that use this kind of reasoning on events (i.e. temporal relations) to improve the predictions.
>
> Thank you for the pointers to these works. Both GTRM [1] (cited as [24] in the paper) and CTRN [2] are performing data-driven graph-based reasoning, which is orthogonal to the proposed method. DTL is a framework that incorporates constraints into the data-driven training process of temporal analysis models. It does not add new parameters to the backbone like GTRM [1] or propose a whole-new backbone architecture like CTRN [2] to support reasoning. Therefore, while both methods are interesting and worth more discussion in the paper, a direct comparison would be inapplicable. Both methods, however, can serve as the backbone of DTL like MS-TCN and ASFormer. Unfortunately, their source code is unavailable.
>
> > Also it could be interesting to combine both approaches to see if a priori-knowledge and learnt logical constraints can bring improvement to work that learns events relations directly from the data. How complementary are both approaches?
>
> Thank you for this interesting question. DTL is complementary with other data-driven reasoning methods because it is agnostic of the backbone’s architecture. We can treat a data-driven reasoning module as a part of the backbone and train it with DTL.
> To demonstrate this, we apply DTL on HASR [3], a reasoning module similar to GTRM [1] and whose source code is released (we need source code to rerun experiments on the same hardware [4] so that performance gain is fairly computed).
>
> **Table R8: Performance gain on 50Salads**
> | Model          | Edit | F1@10 | F1@25 | F1@50 |  Acc |
> |----------------|:----:|:-----:|:-----:|:-----:|:----:|
> | GRU+HASR       | 13.8 |  13.0 |  14.1 |  13.0 | -0.4 |
> | GRU+HASR+DTL   | 14.3 |  13.5 |  14.4 |  13.3 |  0.3 |
> | MSTCN+HASR     |  5.1 |  5.5  |  5.8  |  5.2  |  0.8 |
> | MSTCN+HASR+DTL |  6.2 |  6.4  |  5.9  |  5.8  |  2.1 |
>
> We would like to clarify that DTL does not aim to downplay or negate the importance of architecture design in temporal reasoning (e.g. advance from GRU to MSTCN, then to the ASFormer). Our point is that DTL can work as a complement to existing temporal models despite their different architectures (i.e. GRU/recurrent, MSTCN/convolutional, and ASFormer/transformer-like).
>
> > It is not clear whether the model can work in case of datasets where different actions can happen at the same time (co-occurrent events) as in Epic-Kitchen?
>
> It is possible to describe constraints about event co-occurrence in temporal logic given its expressiveness and extendability. For example, let us define operator $G$ as $Gx=\neg F(\neg x)$, which is true when event x is true in all time steps. A constraint that enforces co-occurrence of event $a_i$ and $a_j$ can be written as: $G((a_i \land a_j) \lor (\neg a_i \land \neg a_j))$, which reads “in all time steps, $a_i$ and $a_j$ must both occur, or both not occur”. We agree that incorporating those constraints for more complicated cases would be a very interesting part of future research works.
>
> > The logical constraints may be new to computer vision but it has been used for a long time on deep learning and since the contribution they provide is not on computer vision (add a logical loss that was used in other deep learning studies) makes the contribution limited.
>
> We respectfully disagree with the statement that DTL is to “add a logical loss that was used in other deep learning studies”. The temporal logic loss used in DTL is not a duplicate of existing literature. As discussed in Section 2.3, we agree that logic loss for data-driven models has been studied for a long time. However, most of the efforts have been on propositional logic for non-sequential data, and the temporal aspect is not well studied. While existing literature motivated this work, it is non-trivial to design a logic loss that accounts for the temporal constraints for the action analysis problems.
>
> > The proposed method is easy to use in datasets such as breakfast and 50 salades (datasets with fixed scripts). However in the case of Epic-Kitchen, there are more events and more complex actions, where it is more challenging to define event constraints and so results on this dataset can better prove the authors claims.
>
> Thank you for the suggestion. The main claim of this paper is that explicitly enforcing constraints to data-driven models through DTL improves their performance. We believe this claim has been validated by the experiment. Having said that, we agree that the benefit of DTL can be even better shown by experiments on larger datasets with richer types of events and more challenging cross-event relations. It would be a very nice study as we explore more efficient ways to express large-scale temporal constraints with logic formulae. We will add relevant study in the final version of this paper.

---

> > ### Author Response · Authors · 2022-08-02
> > **(continued)**
> >
> > > When the base network can better model the temporal relations, the improvement brought by the proposed approach is less important as it is the case of MSTCN and Asformer compared to GRU.
> >
> > We appreciate the Reviewer for this observation, but it is not a weakness of DTL. In fact, a more advanced backbone model is naturally better at learning temporal constraints. DTL can be completely useless on an ideal model that can learn all the constraints from data. Unfortunately, such a model is nonexistent at present.
> >
> > > The model shows improvement especially in the case of small amounts of training data… When the dataset is bigger (more instances), the improvement is not as important, as we can see on 50Salades and BreakFast.
> >
> > In our paper, the claim is that DTL is effective on datasets with different natures (shorter or longer, single-themed or multi-themed). Breakfast and 50Salads differ in many ways. While Breakfast has more instances and more activity categories, each of its video are shorter (2097 frames/video compared with 11511 frames/video in 50Salads) and contain fewer actions (6.8 actions/video compared with 19.9 actions/video in 50Salads) than 50Salads. Both datasets used in assessment are challenging in different ways and the constraints can work differently on these datasets.
> >
> > ### References
> > [1] Huang, Y., Sugano, Y., & Sato, Y. (2020). Improving action segmentation via graph-based temporal reasoning. In Proceedings of the IEEE/CVF conference on computer vision and pattern recognition (pp. 14024-14034).
> >
> > [2] R. Dai, S. Das and F. Bremond. CTRN: Class Temporal Relational Network For Action Detection. In Proceedings of the 32nd British Machine Vision Conference, BMVC 2021, United Kingdom, Virtual, November 22-25, 2021.
> >
> > [3] Ahn H, Lee D. Refining action segmentation with hierarchical video representations. In Proceedings of the IEEE/CVF International Conference on Computer Vision. 2021: 16302-16310.
> >
> > [4] https://docs.nvidia.com/cuda/cublas/index.html#cublasApi_reproducibility

---

> > ### Comment · Reviewer_TJYF · 2022-08-04
> > **Feedback to authors**
> >
> > The authors have convincingly answered most of the questions and have provided important details about the proposed method. However, I still have two remaining concerns :
> >
> > - Building the temporal constraints: From the authors' explanation of how the curation is performed (in an automated manner), it still looks difficult to build the constraints for epic-kitchen, where the correlation between actions is low and the actions are more complex. This issue could be better discussed, explaining for which datasets the proposed method could be more effective.
> >
> > - Comparison with other temporal reasoning methods:
> > Yes, this is true that CTRN and GTRM are somewhat orthogonal to the proposed method, but they serve the same purpose. Even though there is no code for a direct comparison, the Charades dataset (for instance, used in CTRN) is public. So maybe, the authors could run their method on this dataset. It would also serve as an example for cases, where actions co-occurred at the same time step. In the case of Charades, this co-occurrence happens in many samples.
> > I agree that DTL can work as a complement to these existing temporal models. So, it will be interesting to better describe which temporal constraints the proposed method can add.

---

> > > ### Author Response · Authors · 2022-08-05
> > > **Thank you for the follow-up comments. Please find below our response.**
> > >
> > > > It still looks difficult to build the constraints for epic-kitchen, where the correlation between actions is low and the actions are more complex. This issue could be better discussed, explaining for which datasets the proposed method could be more effective.
> > >
> > > Thank you for the thoughtful comment. This is a limitation of how currently knowledge is utilized in the paper. The constraint curation method in the paper depends on a subset of knowledge of actions (i.e., correlation expressible in the form of frequency matrices). We agree that when such correlation is sparse, the knowledge provides less information, and DTL will be less effective.
> > >
> > > Possible solutions to this include:
> > > - improving backbone architecture to better model the non-correlation knowledge (e.g., visual information) from data.
> > > - improving the curation method to better represent non-correlation knowledge (e.g., with human involvement).
> > >
> > > Still, we would like to highlight that:
> > > - DTL is useful for non-correlation temporal knowledge as long as it can be written as temporal logic formulae.
> > > - Correlation knowledge plays an important role when the video is about an activity made of a series of logically related actions.
> > >
> > > Epic-Kitchen differs from any of the datasets we have discussed here because its actions are verb-object pairs, which require two sets of constraints: (1) the affordance constraints on verb-object composition and (2) the temporal constraints similar to what is described in this paper. A comprehensive study of this dataset will be a very exciting assessment of the expressiveness of logic for this task. We will add discussion on this aspect.
> > >
> > > > The authors could run their model on Charades, where actions co-occurred at the same time step. It will be interesting to better describe which temporal constraints the proposed method can add.
> > >
> > > While we believe the provided experiments sufficiently support the claims of our paper, we agree that it is an important and interesting extension to assess DTL on Charades and Epic-Kitchen, where multiple actions co-occur simultaneously. Therefore, following the Reviewer’s comment, we will examine Charades and provide studies in the final version of this paper, and we believe it will add value to this research.
> > >
> > > We can provide some early statistics about the constraints we curated on Charades: we have curated 9668 constraints, of which 2,848 are Backward Dependency, 2,848 are Forward Cancellation, 3,972 are Exclusivity, and none are Implication constraints.
> > >
> > > As a side note, constraints in the paper are effective whether or not simultaneous co-occurrence is allowed. This is because they only regulate the order (by BD constraints) and presence of actions (FC, Ex, and Ip constraints).

---

### Official Review · Reviewer_7jYQ · 2022-07-09

**Rating:** 7
**Confidence:** 4
**Soundness:** 3 good
**Presentation:** 4 excellent
**Contribution:** 3 good

**Summary:**

This paper proposed differential temporal logic to model temporal dependencies, such as co-occurrence and ordering. Based on logic operators, four constraints were defined, including backward dependency, forward cancellation, implication and exclusivity. The proposed method is evaluated on two datasets of the temporal action segmentation task.

**Questions:**

Please see "weaknesses".

**Limitations:**

The limitations were briefly discussed in the paper, while the potential negative societal impact was not.

**Strengths And Weaknesses:**

Strengths:

1. The paper is well-motivated and well-written. Modeling temporal dependencies are an important contribution for temporal action segmentation.

2. The proposed differential temporal logic is very interesting. This method is principled and extendable to include more constraints.

3. Effectiveness is shown on two datasets.

Weaknesses:

1. [Critical] How the constraints are curated from the datasets is not clear enough.
- The curation process need to be described more.
- Are they curated from the training split or the whole dataset? If they are curated from the training split, there should have been k sets of constraints instead of a single set as in the supplementary.
- Will the curation codes be released?

2. Why the performance of ASFormer reported in this paper is different from the original ASFormer paper?

3. I would recommend add ablation studies on the hyper-parameter \lamda and \gamma to see their sensitivities.

4. How the examples are selected for Fig.5 and Fig.6. Are they representative? What is the model g for Fig.6? From Fig.6, we could see that the Ip constraints are much stronger than others, while in Table 3 the gain from Ip constraints is not as stronger than others. Please discuss this.

Recommandations:

1. It would be interesting if the proposed method could be extended to more constraints, such as:
- the co-occurrence at the same time step
- combination of actions in the constraints (e.g., action a happens only if action b and action c have both happened before)

2. How many times are the experiments repeated?

---

> ### Author Response · Authors · 2022-08-02
> **Thank you very much for the comments. In summary, below are our responses regarding the curation of constraints, the performance of baselines, and other issues about experiments.**
>
> > The curation process needs to be described more.
>
> Since Reviewer Zn5F raised a similar issue, please check Message to All Reviewers for our response. We commit ourselves to include the discussion in revision.
>
> > Are they curated from the training split or the whole dataset?
>
> The constraints are curated from all the annotations (so “training annotations” in line 173 should be corrected to “annotations”). In general, the knowledge expressed as constraints in DTL should cover as much as possible the domain knowledge, which includes the knowledge in different parts of the annotation.
>
> We compared the difference between the constraints collected from the whole datasets and from each training split, and found that: (a) the number of different constraints is small, and (b) such difference does not affect the conclusion of experiments. The tables below provide statistics about the difference, and the outcome of experiments run with split-specific constraints.
>
> Denote as $C_0$ the constraints curated from all the annotations and $C_k$ the constraints curated only from the kth split’s training annotations. In the two tables below, the center column is $d_k = |C_k \setminus C_0| + |C_0 \setminus C_k|$, the number of different constraints between $C_k$ and $C_0$. The rightmost column is $d_k / |C_0|$, the percentage of difference.
>
> **Table R1: for 50Salads: $k=1,2,3,4,5$ and $|C_0| = 313$**
> | Split Number | $d_k$ | $d_k / \|C_0\|$ (%) |
> |--------------|:-----:|:---------------:|
> | 1            |   71  |       22.6      |
> | 2            |   0   |        0        |
> | 3            |   8   |       2.5       |
> | 4            |   4   |       1.3       |
> | 5            |   0   |        0        |
>
> **Table R2: for Breakfast: $k=1,2,3,4$ and $|C_0| = 2145$**
> | Split Number | $d_k$ | $d_k / \|C_0\|$ (%) |
> |--------------|:-----:|:---------------:|
> | 1            |   37  |       1.7       |
> | 2            |   38  |       1.8       |
> | 3            |   28  |       1.3       |
> | 4            |   34  |       1.6       |
>
> We also rerun the experiments in Table 1 and Table 2 with split-only constraints. The two tables below show that such difference has limited impact on the performance gain, and does not affect the conclusion of experiments. In the two tables below, settings with “all” use all the annotations as in the paper. Settings with “split” use constraints from each split’s training annotations.
>
> **Table R3: 50Salads: comparison between constraint from all annotations and split annotations**
> | Gain                 | Edit | F1@10 | F1@25 | F1@50 | Acc |
> |----------------------|:----:|:-----:|:-----:|:-----:|:---:|
> | GRU+DTL (all)        |  7.0 |  5.4  |  6.0  |  5.9  | 1.5 |
> | GRU+DTL (split)      |  7.1 |  5.6  |  6.0  |  5.5  | 1.3 |
> | MSTCN+DTL (all)      |  2.1 |  3.6  |  3.5  |  2.3  | 0.8 |
> | MSTCN+DTL (split)    |  1.0 |  2.6  |  3.5  |  3.0  | 1.5 |
> | ASFormer+DTL (all)   |  3.8 |  3.9  |  4.3  |  5.0  | 2.9 |
> | ASFormer+DTL (split) |  3.6 |  3.5  |  4.2  |  4.6  | 2.7 |
>
> **Table R4: Breakfast: comparison between constraint from all annotations and split annotations**
> | Gain              | Edit | F1@10 | F1@25 | F1@50 | Acc |
> |-------------------|:----:|:-----:|:-----:|:-----:|:---:|
> | GRU+DTL (all)     |  1.8 |  3.6  |  3.1  |  2.4  | 0.3 |
> | GRU+DTL (split)   |  1.6 |  3.2  |  3.0  |  2.3  | 0.4 |
> | MSTCN+DTL (all)   |  0.4 |  1.3  |  2.1  |  2.4  | 0.9 |
> | MSTCN+DTL (split) |  0.5 |  1.2  |  2.0  |  2.1  | 1.1 |
>
> We expect similar small changes for ASFormer on Breakfast. Given the time constraint, however, we are unable to provide the results here.
>
> > Will the curation codes be released?
>
> Yes, we will release all the source code, including the code that generates the temporal logic formulae, as promised in the paper.
>
> > Why the performance of ASFormer reported is different from the original ASFormer paper?
>
> To avoid overestimating/underestimating the performance gain, we need to retrain all the baseline models to compare with baseline+DTL using the same hardware. This is because exact reproducibility is not guaranteed for CUDA across GPU hardware [1]. For example, for MS-TCN on Breakfast, we obtained a higher baseline performance than the original paper. In this case, using performance from the original paper will cause an overestimation of performance gain. We will add the performance from the original papers for completeness.

---

> > ### Author Response · Authors · 2022-08-02
> > **(continued)**
> >
> > > I would recommend add ablation studies on the hyper-parameter $\lambda$ and $\gamma$ to see their sensitivities.
> >
> > Thank you for the suggestion. We performed ablation studies on MS-TCN and GRU on 50Salads using performance gain on F1@25 as the metric. DTL is more sensitive to $\lambda$ than $\gamma$. A too large weight on temporal constraints can negatively affect the training because it essentially requires the model to focus more on temporal constraints than on the training data.
> >
> > **Table R5: F1@25 performance gain of DTL on GRU or MSTCN with different $\lambda$**
> > | $\lambda$ | 0.01 | 0.05 | 0.1 | 0.2 |  0.5 |
> > |-----------|:----:|:----:|:---:|:---:|:----:|
> > |  GRU      |  2.4 |  4.7 | 6.0 | 2.5 | -3.2 |
> > |  MSTCN    |  1.0 |  2.0 | 3.5 | 2.2 |  0.5 |
> >
> > **Table R6: F1@25 performance gain of DTL on GRU or MSTCN with different $\gamma$**
> > | $\gamma$ | 0.5 | 1.0 | 10.0 |
> > |----------|:---:|:---:|:----:|
> > |  GRU     | 6.7 | 6.0 |  5.6 |
> > |  MSTCN   | 3.1 | 3.5 |  1.2 |
> >
> > > How the examples are selected for Fig.5 and Fig.6. Are they representative?
> >
> > They are manually selected to show the efficacy of DTL and are representative. To show this (i.e. DTL can improve backbones to make its output consistent with constraints), we compare the percentage of satisfied constraints of backbone-only models and backbone+DTL on the test set. The results below show that backbone+DTL satisfy more constraints:
> >
> > **Table R7: Average percent of constraints that the model output satisfies on the test set**
> > | Model        | 50Salads | Breakfast |
> > |--------------|:--------:|:---------:|
> > | GRU          |   70.4%  |   77.2%   |
> > | GRU+DTL      |   93.2%  |   100.0%  |
> > | MSTCN        |   69.4%  |   61.3%   |
> > | MSTCN+DTL    |   89.2%  |   99.4%   |
> > | ASFormer     |   68.9%  |   40.7%   |
> > | ASFormer+DTL |   84.8%  |   66.3%   |
> >
> > > What is the model g for Fig.6? Please discuss more on Fig. 6.
> >
> > We used MSTCN as the backbone $g$ for Figure 6. Figure 6 is not showing the strength of different types of constraints. Instead, it shows the effect of different constraints on the logits for different actions. The greenish color for Ip constraints means that Ip constraints tend to increase the logits of actions (“promotive” as in the paper because it encourages actions to occur). On the other hand, the blue-ish color for Ex, FC, and BD constraints means that these three types of constraints tend to decrease the logits of actions (“suppressive” as in the paper).
> >
> > > It would be interesting if the proposed method could be extended to more constraints.
> >
> > We agree that there are many constraints that can benefit DTL in more complicated domains. The two suggested cases can be incorporated in DTL framework as follows:
> >
> > - Co-occurrence of $a_i$ and $a_j$: $G((a_i \land a_j) \lor (\neg a_i \land \neg a_j))$, which reads “in all time steps, $a_i$ and $a_j$ must both occur, or both not occur”. Here, $G x=\neg F(\neg x)$ is true when event x is true in all time steps.
> > - Combinations of more actions: $F(a_i \land a_j \land a_k) \rightarrow ((\neg a_i W a_j) \land (\neg a_i W a_k))$, which is an extended backward dependency constraint that says $a_i$ is dependent on $a_j$ and $a_k$.
> >
> > > How many times are the experiments repeated?
> >
> > The k-fold cross-validation experiments were performed once with the seed set to 0 for all the experiments for a fair comparison, where $k=5$ for 50Salads and $k=4$ for Breakfast following the protocol in[14][53].
> > The error bars reported in Table 1 and Table 2 are from cross-validation.
> >
> > ### References
> > [1] https://docs.nvidia.com/cuda/cublas/index.html#cublasApi_reproducibility

---

> > ### Comment · Reviewer_7jYQ · 2022-08-07
> > **Feedback to Authors**
> >
> > Thank you very much for the comprehensive response.
> >
> > I think curating constraints from only the training annotations is the correct way here, since curating from all the annotations would leak information in the testing splits. Given that the performance gains are not affected evidently, I would not consider this as a significant issue. However, please update all the experiment results to the correct setting in your next version.
> >
> > As for other concerns, they are well addressed. I do like this paper since I found how to model action relations important when I was working on this task, and this paper presents an interesting solution.

---

> > > ### Author Response · Authors · 2022-08-08
> > > **Thank you for the follow-up comment**
> > >
> > > Thank you for the follow-up comment and for recognizing our work. Following the comment, we will update the results under the correct setting in future revisions.

---

### Official Review · Reviewer_Zn5F · 2022-07-10

**Rating:** 5
**Confidence:** 4
**Soundness:** 3 good
**Presentation:** 3 good
**Contribution:** 3 good

**Summary:**

This submission proposes to place temporal logic for temporal action segmentation model training. Compared to previous methods, the authors leverage a Linear Temporal Logic (LTL) formula to evaluate model's prediction and place a new loss item during the training. Experiments show that the proposed methods achieves improvement in performance on two datasets (50Salads and Breakfast datasets).

**Questions:**

There two problems for the temporal logic loss:

1. The collected temporal constraints already provided external knowledge (similar to labeling or supervision) during the training. It may be natural to expect there will be improvement in model's performance.

2. The novelty of the submission lies in that the LTL loss can assist most existing models in temporal action segmentation task. However, the generation of the temporal constraints may not be free and is not scalable. For example, in 220-223, the authors mentioned there are 313 and 2145 constraints on the 50salads and breakfast datasets. It would be better to reveal more details of the generation process (e.g., is there an automatical way to generate such constraints, or we may need to manually select different concepts and explicitly place several constraints as explained in section 3.3 and section E in appendix).

**Limitations:**

Please check the "questions" section above.

**Strengths And Weaknesses:**

+ The motivation and technical details are clear to understand. The authors provide theoretical analysis for the proposed LTL evaluation system during training.

+ The authors provided ablation studies on the two datasets (50Salads and breakfast) in terms of different existing models, which is appreciated.

+ The authors provided the collected temporal constraints on these two datasets, which can further assist future research.

---

> ### Author Response · Authors · 2022-08-02
> **Thank you very much for the comments. In summary, below are our responses regarding the improvement of performance and the curation of constraints.**
>
> Thank you very much for the comments. In summary, below are our responses regarding the improvement of performance and the curation of constraints.
>
> > The collected temporal constraints already provided external knowledge (similar to labeling or supervision) during the training. It may be natural to expect there will be improvement in model's performance.
>
> The constraints are extracted from the existing annotations, so strictly speaking we did not introduce extra knowledge but explicitly presented existing knowledge to the model. We agree that improvement is natural when the knowledge in the training dataset (in the form of annotation) can be better learned by the model. This is in fact the motivation of the proposed method – we provide a way that
>
> 1. explicitly enforces those constraints in case the backbone model fails to capture them from data, and
>
> 2. is compatible with most of the end-to-end trainable models (in the form of logic loss), so our method is complementary to the existing merits of those backbones.
>
> > The generation of the temporal constraints may not be free and is not scalable. It would be better to reveal more details of the generation process.
>
> The constraints are extracted from the existing annotation in the dataset and does not require manual effort. Please check Message to All Reviewers for the details of this process. We commit ourselves to including the discussion in revision.
>
> > Is there an automatic way to generate such constraints, or we may need to manually select different concepts and explicitly place several constraints as explained in section 3.3 and section E in appendix.
>
> Collecting knowledge, converting it to constraints, and enforcing it via DTL is automatic as long as the types of knowledge are defined. It would be exciting yet challenging to design more general methods to curate free-form knowledge/constraints in future study. We are looking forward to combining those methods with the DTL framework.

---

### Author Response · Authors · 2022-08-02
**Message to All Reviewers**

We would like to thank all the reviewers for their time and efforts in providing feedback on our paper. The responses to each reviewer’s comments are posted under the corresponding thread.

We would like to detail the curation of constraints in response to the common issue raised by Reviewers Zn5F and 7jYQ. The curation process is automatic and does not require manual involvement.

For an input sample of $T$ frames, its annotation can be written as a sequence $[y_1, y_2, …, y_T]$, where $y_i \in {0, 1, …, N-1}$ is the index for one of the $N$ action categories. We assume we have $M$ such samples, and only one action can occur given any time step.

The curation process starts with the collection of the following statistics:
- B: `B[a_i, a_j]` is the frequency of action $a_i$ occurring before $a_j$
- P: `P[a_i, a_j]` is the frequency of action $a_i$ occurring after $a_j$
- J: `V[a_i, a_j]` is the number of videos where $a_i$ and $a_j$ occur (but not simultaneously)
- C: `C[a_i]` is the number of videos where $a_i$ occurs

The collection procedure is described as the pseudo-code below:
```
B, P, J <- zero matrices of size NxN
C <- zero vectors of size N
for sample m in M:
    occur_flags <- zero vectors of size N
    co_occur_flags <- zero matrix of size NxN
    y_1, y_2, ..., y_T <- annotation of sample m
    for t in 1, 2, ..., T:
        if occur_flags[y_t] == 0:
            C[y_t] <- C[y_t] + 1
            occur_flags[y_t] <- 1
        for u in 1, 2, ..., t-1:
            B[y_u, y_t] <- B[y_t’, y_t] + 1
            if co_occur_flags[y_t, y_u] == 0:
                J[y_u, y_t] <- J[y_u, y_t] + 1
                co_occur_flags[y_u, y_t] = 1
        for u in t+1, t+2, ..., T:
            P[y_u, y_t] <- P[y_u, y_t] + 1
            if co_occur_flags[y_t, y_u] == 0:
                J[y_u, y_t] <- J[y_u, y_t] + 1
                co_occur_flags[y_u, y_t] = 1
```
Then we generate the constraints as follows:
```
for i in 0, 1, 2, ..., N-1:
    for j in 0, 1, 2, ..., N-1:
        if i != j and J[i,j] > 0 and B[i,j] == 0:
            append_BD(i, j)  # action i is “backward dependent” on j
        if J[i,j] > 0 and P[i,j] == 0:
            append_FC(j, i)  # action j “forward cancels” j
        if i != j and J[i,j] / C[j] == 1:
            append_Ip(j, i)  # action j implies i
        if i != j and J[i,j] == 0:
            append_Ex(i, j)  # action i and j is exclusive
```

---

### Meta-Review · Area_Chair_DRrG · 2022-08-27

**Recommendation:** Accept
**Confidence:** Certain

**Metareview:**

This paper introduces an approach for incorporating declarative temporal constraints in the training of temporal action segmentation models, in a model-agnostic fashion. Reviewers generally appreciated the proposed approach, but questioned the scalability and generalizability of the constraint curation process and asked for experimental results on more complex and challenging datasets beyond 50Salads and Breakfast. The author responses addressed many of the reviewer concerns, and reviewers responded positively overall. However, not all concerns could be addressed within the rebuttal time; for example, the authors promised to add results on more complex datasets such as Epic-Kitchen and Charades where reviewers were concerned it may be more challenging to generate constraints. After reading the paper and all author and reviewer responses, I believe the contributions of the paper are sufficient for acceptance. However, authors are expected to add the experimental results on additional datasets as promised for the final paper. Baseline numbers from original papers for some of the models that needed to be re-implemented should also be included for comparability.

**Award:**

No

---

### Decision · Program_Chairs · 2022-09-14

Accept